# Therapeutic Switching of Rafoxanide: a New Approach To Fighting Drug-Resistant Bacteria and Fungi

Mahmoud M. Bendary,[a] Marwa I. Abd El-Hamid,[b] Amira I. Abousaty,[c] Arwa R. Elmanakhly,[d] Walaa A. Alshareef,[e] Rasha A. Mosbah,[f] Majid Alhomrani,[g,h] Mohammed M. Ghoneim,[i] Amr Elkelish,[j,k] Nada Hashim,[l] Abdulhakeem S. Alamri,[g,h] Helal F. Al-Harthi,[m] Nesreen A. Safwat[d]

[a]Department of Microbiology and Immunology, Faculty of Pharmacy, Port Said University, Port Said, Egypt

[b]Department of Microbiology, Faculty of Veterinary Medicine, Zagazig University, Zagazig, Egypt

[c]Department of Microbiology, Faculty of Science, Zagazig University, Zagazig, Egypt

[d]Department of Microbiology and Immunology, Faculty of Pharmacy, Modern University for Technology and Information, Cairo, Egypt

[e]Department of Microbiology and Immunology, Faculty of Pharmacy, October 6 University, 6th of October, Egypt

[f]Infection Control Unit, Zagazig University Hospital, Zagazig, Egypt

[g]Department of Clinical Laboratories Sciences, The Faculty of Applied Medical Science, Taif University, Taif, Saudi Arabia

[h]Centre of Biomedical Science Research, Deanship of Scientific Research, Taif University, Taif, Saudi Arabia

[i]Department of Pharmacy Practice, College of Pharmacy, Al Maarefa University, Ad Diriyah, Saudi Arabia

[j]Biology Department, College of Science, Imam Mohammad ibn Saud Islamic University, Riyadh, Saudi Arabia

[k]Botany and Microbiology Department, Faculty of Science, Suez Canal University, Ismailia, Egypt

[l]Faculty of Medicine, University of Gezira, Wad Medani, Sudan

[m]Biology Department, Turabah University College, Taif University, Taif, Saudi Arabia

Mahmoud M. Bendary and Marwa I. Abd El-Hamid contributed equally to this work. The order was determined by the corresponding author after a mutual agreement among the authors.

**ABSTRACT** Control and management of life-threatening bacterial and fungal infections are a global health challenge. Despite advances in antimicrobial therapies, treatment failures for resistant bacterial and fungal infections continue to increase. We aimed to repurpose the anthelmintic drug rafoxanide for use with existing therapeutic drugs to increase the possibility of better managing infection and decrease treatment failures. For this purpose, we evaluated the antibacterial and antifungal potential of rafoxanide. Notably, 70% (70/100) of bacterial isolates showed multidrug resistance (MDR) patterns, with higher prevalence among human isolates (73.5% [50/68]) than animal ones (62.5% [20/32]). Moreover, 22 fungal isolates (88%) were MDR and were more prevalent among animal (88.9%) than human (87.5%) sources. We observed alarming MDR patterns among bacterial isolates, i.e., *Klebsiella pneumoniae* (75% [30/40; 8 animal and 22 human]) and *Escherichia coli* (66% [40/60; 12 animal and 28 human]), and fungal isolates, i.e., *Candida albicans* (86.7% [13/15; 4 animal and 9 human]) and *Aspergillus fumigatus* (90% [9/10; 4 animal and 5 human]), that were resistant to at least one agent in three or more different antimicrobial classes. Rafoxanide had antibacterial and antifungal activities, with minimal inhibitory concentration (MICs) ranging from 2 to 128 $\mu$g/mL. Rafoxanide at sub-MICs downregulated the mRNA expression of resistance genes, including *E. coli* and *K. pneumoniae* $bla_{CTX-M-1}$, $bla_{TEM-1}$, $bla_{SHV}$, *MOX*, and *DHA*, *C. albicans ERG11*, and *A. fumigatus cyp51A*. We noted the improvement in the activity of $\beta$-lactam and antifungal drugs upon combination with rafoxanide. This was apparent in the reduction in the MICs of cefotaxime and fluconazole when these drugs were combined with sub-MIC levels of rafoxanide. There was obvious synergism between rafoxanide and cefotaxime against all *E. coli* and *K. pneumoniae* isolates (fractional inhibitory concentration index [FICI] values $\leq$ 0.5). Accordingly, there was a shift in the patterns of resistance of 16.7% of *E. coli* and 22.5% of *K. pneumoniae* isolates to cefotaxime and those of 63.2% of *C. albicans* and *A. fumigatus* isolates to fluconazole when the isolates were treated with sub-MICs of rafoxanide. These results were confirmed by *in silico* and mouse

Address correspondence to Nada Hashim, med@uofg.edu.sd, or Mahmoud M. Bendary, micro_bendary@yahoo.com.

The authors declare no conflict of interest.

protection assays. Based on the *in silico* study, one possible explanation for how rafoxanide reduced bacterial resistance is through its inhibitory effects on bacterial and fungal histidine kinase enzymes. In short, rafoxanide exhibited promising results in overcoming bacterial and fungal drug resistance.

**IMPORTANCE** The drug repurposing strategy is an alternative approach to reducing drug development timelines with low cost, especially during outbreaks of disease caused by drug-resistant pathogens. Rafoxanide can disrupt the abilities of bacterial and fungal cells to adapt to stress conditions. The coadministration of antibiotics with rafoxanide can prevent the failure of treatment of both resistant bacteria and fungi, as the resistant pathogens could be made sensitive upon treatment with rafoxanide. From our findings, we anticipate that pharmaceutical companies will be able to utilize new combinations against resistant pathogens.

**KEYWORDS** repurposing, rafoxanide, resistance, treatment failure, *in silico*, mouse protection, resistance gene downregulation

Animals and human are prone to serious bacterial and fungal infections. The significance of these infections has increased dramatically in recent decades. More importantly, several bacterial and fungal species have adopted specialized approaches to avoid the inhibitory effects of antimicrobials, with multidrug-resistant (MDR) pathogens being discovered at an alarming frequency. Infections caused by MDR pathogens are progressively common and represent serious and ever-increasing threats to public health, particularly in terms of adverse clinical outcomes (1). Currently, there is a tendency for pathogens to shift from MDR to extensively drug resistant or probably pan-drug resistant, implying substantial risks (2, 3).

Clinically, the misuse and overuse of $\beta$-lactam and other antifungal drugs are crucial factors promoting antibiotic resistance, which accounts for the loss of their effectiveness, treatment failure and worse infection outcomes (4). This crisis has been compounded by the emergence of newer $\beta$-lactamase enzymes that destroy the $\beta$-lactam ring in the $\beta$-lactam drugs, including extended-spectrum $\beta$-lactamases (ESBLs) such as temoneira (TEM), cefotaximase (CTX) and, the most predominant, sulfhydryl variable (SHV) and AmpC $\beta$-lactamases (5). The latter enzymes are encoded on either chromosomes or plasmids, with *MOX* and *DHA* being the most important AmpC $\beta$-lactamase-encoding genes (6).

Concerning fungal infections, the selection of unique fungal targets for antimicrobial therapies represents a great challenge owing to the similarity between fungal and human cells (7). Moreover, several studies reported that most fungal infections were attributable to pathogens with MDR phenotypes, especially in immunocompromised patients, with the consequence that a broad range of antifungals are ineffective (8). Resistance to azole antifungal drugs is common among *Candida* and *Aspergillus* species. Azole resistance is associated with the overexpression of *C. albicans ERG11* (encoding lanosterol demethylase) and *A. fumigatus cyp51* (encoding sterol 14 alpha-demethylase) (9, 10).

Many researchers have provided new insights into critical approaches that would overcome the antimicrobial resistance issues, such as using natural alternative and complementary therapies (11) and drug repurposing (12). The natural therapies have several disadvantages, as they might interact with other prescribed therapies and they are contraindicated in several cases (13). For that reason, drug repurposing is an effective strategy developed for identifying new therapeutic uses for existing therapies. The halogenated salicylanilide *N*-[3-chloro-4-(4-chlorophenoxy)phenyl]-2-hydroxy-3,5-diiodobenzamide (rafoxanide), a veterinary salicylanilide anthelmintic drug with high activity against multiple nematode species (14), can be repurposed for use against microbial infections.

Obviously, there is a difference in the antimicrobial and antiparasitic mechanisms of rafoxanide. The antiparasitic activity is attributed to interference with energy metabolism and a major role in chitinase inhibition. Rafoxanide can inhibit the helminths' ATP production, especially hematophagous helminths, as it is an uncoupler of the mitochondrial oxidative phosphorylation (15). This drug has high affinity for binding with

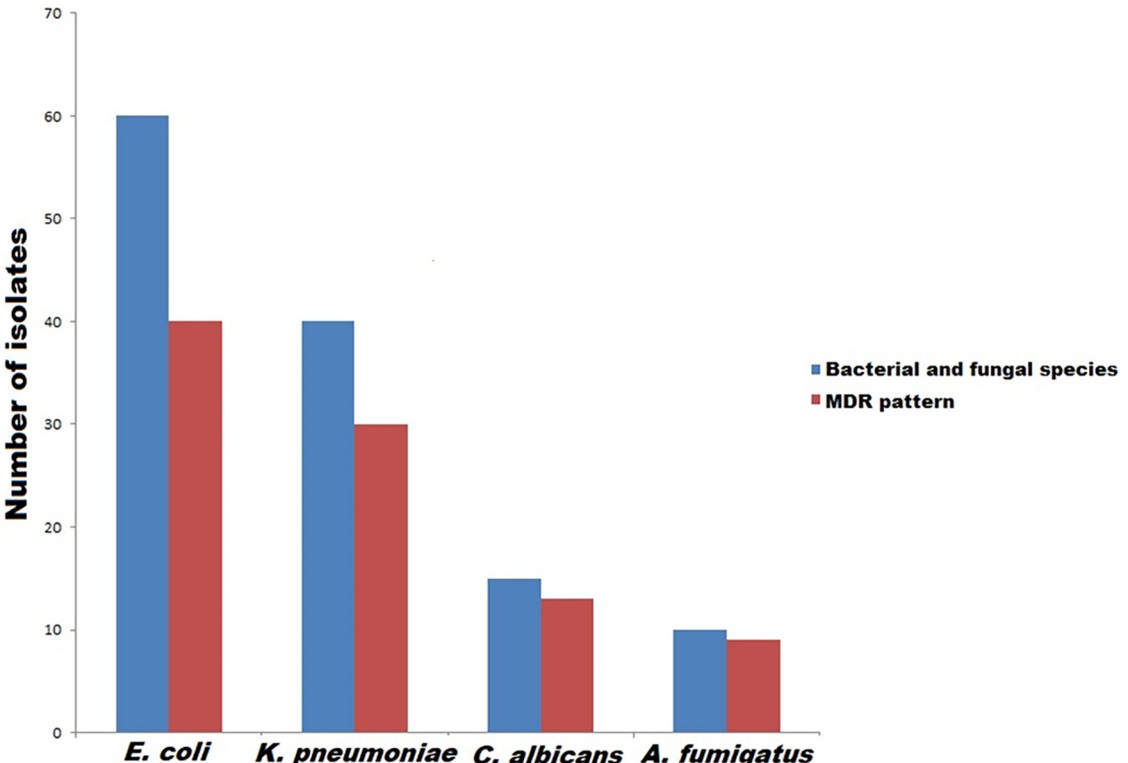

**FIG 1** Numbers of bacterial (*E. coli* and *K. pneumoniae*) and fungal (*C. albicans* and *A. fumigatus*) isolates recovered in the current study and those exhibiting multidrug resistance patterns. MDR is defined as resistance to at least one antimicrobial agent in three or more different antimicrobial classes. The MDR percentage was calculated in relation to the total number of recovered isolates from each bacterial and fungal species.

albumin and surface protein on red blood cells (RBCs), and it persists in blood until the helminths start to consume the blood. When the concentration of salicylanilide derivatives reaches its minimal inhibitory concentration (MIC), it can act on the helminths' mitochondria (16). This mechanism cannot be applied when the therapeutic use of this drug is switched to microbial cells. Of note, salicylanilide derivatives as antimicrobial therapies have several primary targets and multiple antimicrobial mechanisms, such as their direct action on cell membrane permeability; however, they do not disrupt this membrane, and so this activity cannot be considered a sole or central mechanism (17).

It was shown previously that drugs containing a salicylanilide nucleus, such as closantel, have significant inhibitory effects on the two-component signaling systems (TCSs) of *Bacillus subtilis* and *Escherichia coli*. This system allows bacteria and fungi to adapt and survive in the presence of antimicrobial agents (18, 19) and is composed of a sensor histidine kinase (HK) and a response regulator (RR) (20), suggesting important implications for developing novel TCS inhibitors (21).

Therefore, this study was carried out to repurpose the anthelmintic drug rafoxanide as a novel strategy to reduce antimicrobial resistance in both bacterial (*E. coli* and *Klebsiella pneumoniae*) and fungal (*Candida albicans* and *Aspergillus fumigatus*) pathogens.

## RESULTS

**Characterization of bacterial and fungal isolates.** On the basis of phenotypic identification data, 60 *E. coli* (20 animal and 40 human) and 40 *K. pneumoniae* (12 animal and 28 human) isolates were recovered from 200 diarrhetic samples with prevalence rates of 30 and 20%, respectively (Fig. 1). In parallel with phenotypic detection results, *E. coli* and *K. pneumoniae* isolates were confirmed via an API 20E system and PCR amplification of *16S* rRNA and 16S-23S internal transcribed spacer genes, respectively. Of 100 ear swabs, 16/50 (32%) and 9/50 (18%) fungal growths were recorded from patients and dogs, respectively, with an overall prevalence rate of 25%. The fungal isolates were characterized

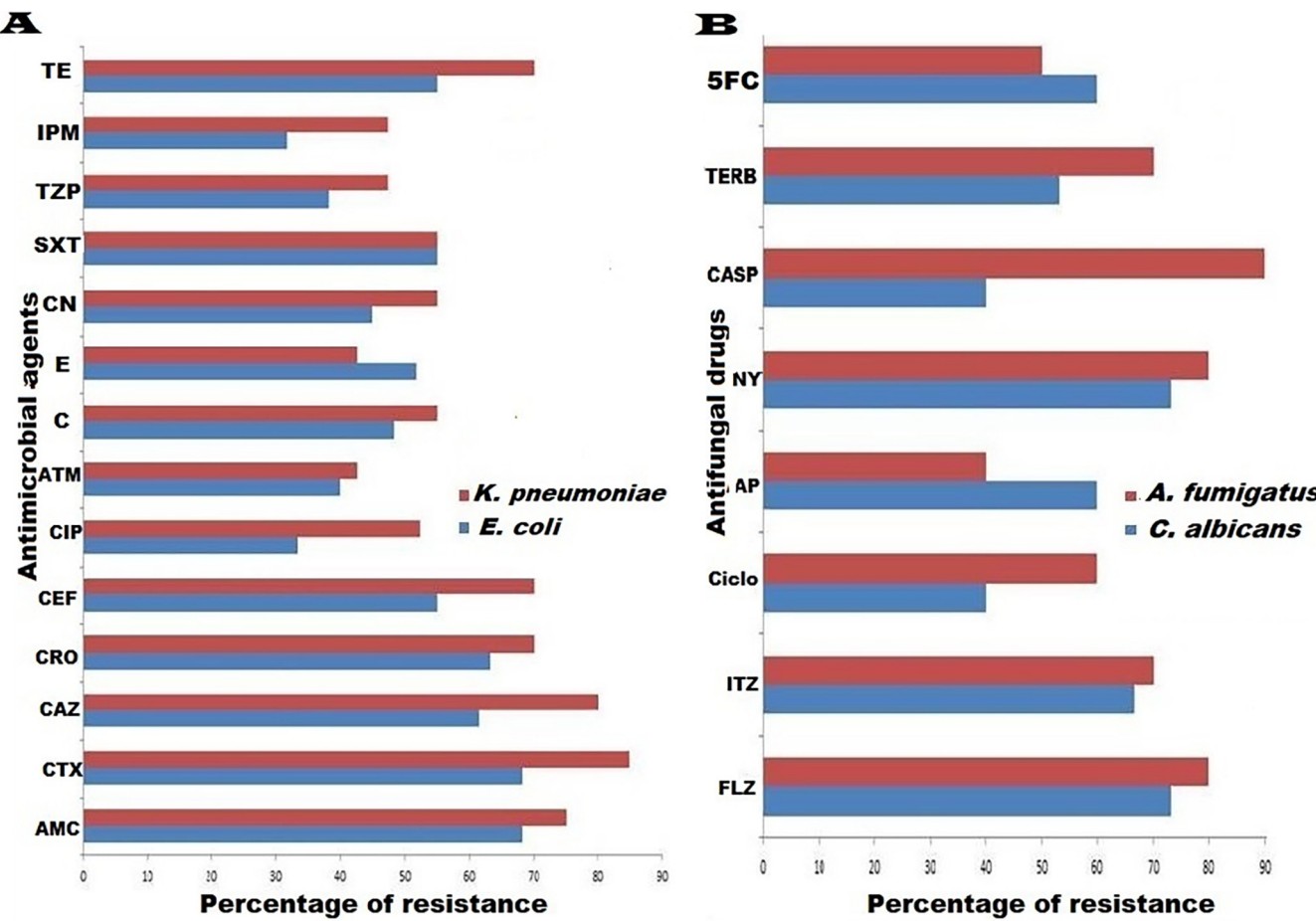

**FIG 2** Frequency of resistance to antimicrobials among all recovered *E. coli* (*n* = 60), *K. pneumoniae* (*n* = 40), *C. albicans* (*n* = 15), and *A. fumigatus* (*n* = 10) isolates based on Kirby-Bauer disc diffusion and broth microdilution methods using Mueller-Hinton agar and commercial antimicrobial and antifungal disks. Inoculum preparation and data interpretation were conducted in accordance with CLSI recommendations. Both procedures were performed in three biological replicates. AMC, amoxicillin-clavulanic acid; CTX, cefotaxime; CAZ, ceftazidime; CRO, ceftriaxone; CEF, cefepime; CIP, ciprofloxacin; ATM, aztreonam; C, chloramphenicol; E, erythromycin; CN, gentamicin; SXT, trimethoprim-sulfamethoxazole; TZP, piperacillin-tazobactam; IPM, imipenem; TE, tetracycline; FLZ, fluconazole; ITZ, itraconazole; Ciclo, ciclopirox; AP, amphotericin B; NY, nystatin; CASP, caspofungin; TERB, terbinafine; 5FC, 5-fluorocytosine.

at the genus level using phenotype identification characteristics. Among the recovered 25 isolates, 10 *A. fumigatus* (4 animal and 6 human) and 15 *C. albicans* (5 animal and 10 human) isolates were identified (Fig. 1). The species-level identification of the recovered isolates was confirmed according to sequencing of *18S* rRNA PCR products with the GenBank accession numbers OL684561 to OL684585.

**Antimicrobial susceptibility patterns.** Our results revealed high percentages of MDR bacterial and fungal isolates (70% [70/100] and 88% [22/25]), respectively. Of note, MDR patterns were more prevalent among human bacterial isolates (73.5% [50/68]) than animal ones (62.5% [20/32]), but they were higher among animal fungal isolates (88.9% [8/9]) than human ones (87.5% [14/16]). Regarding the distribution of MDR patterns among bacterial and fungal species, they were identified among *K. pneumoniae* (75 [30/ 40; 8 animal and 22 human]), *E. coli* (66 [40/60; 12 animal and 28 human]), *C. albicans* (86.7 [13/15; 4 animal and 9 human]), and *A. fumigatus* (90 [9/10; 4 animal and 5 human]) isolates (Fig. 1). With the exception of erythromycin, *K. pneumoniae* isolates showed higher rates of resistance to all tested antimicrobial agents than *E. coli* ones (Fig. 2A). While high levels of resistance to cefotaxime and ceftazidime were detected in *K. pneumoniae*, low levels of resistance to aztreonam and erythromycin were detected. Regarding *E. coli* isolates, the lowest and highest resistance rates were for imipenem and for amoxicillin-clavulanic acid and cefotaxime, respectively (Fig. 2A). Moreover, with the exception of 5-fluorocytosine and amphotericin B, *A. fumigatus* isolates showed higher rates of resistance to all antifungal drugs than *C. albicans* ones (Fig. 2B and 3). Interestingly, the most effective drugs against

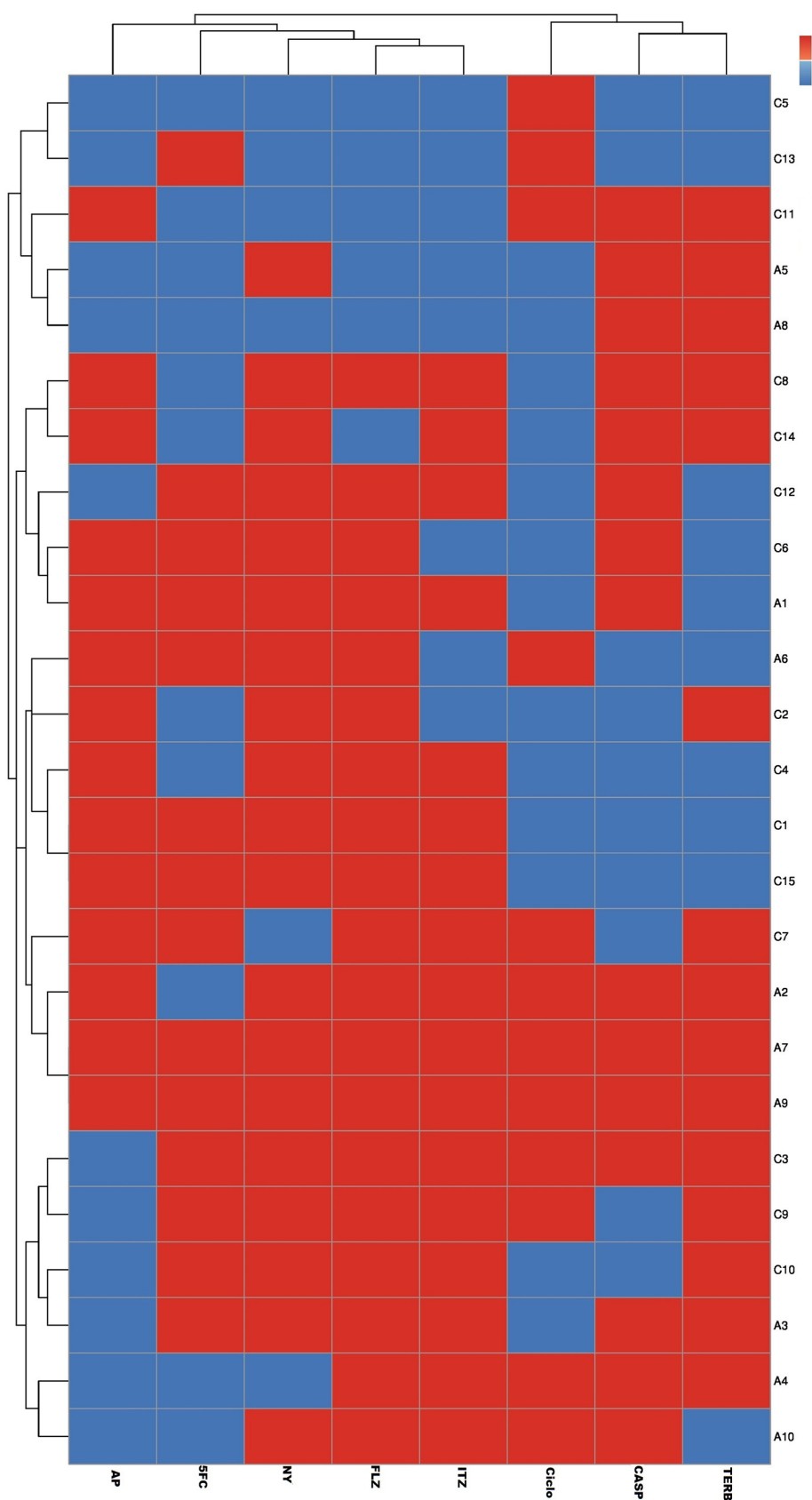

**FIG 3** Heat map showing hierarchical clustering and overall distribution of the investigated *C. albicans* and *A. fumigatus* isolates based on their antifungal resistance patterns. The presence and absence of resistance

*C. albicans* were ciclopirox and caspofungin, with a susceptibility of 60% to each. The amphotericin B was the most suitable antifungal drug against *A. fumigatus* with the lowest resistance rate (40%). In the same context, *C. albicans* and *A. fumigatus* showed higher rates of resistance to both fluconazole and nystatin (73.3% each) and caspofungin (90%) (Fig. 2B and 3). Notably, resistant *C. albicans* (11/15 [73.3%]) and *A. fumigatus* (8/10 [80%]) isolates showed fluconazole MICs of ≥8 and ≥4 $\mu$g/mL, respectively (see Table S1 in the supplemental material). With the exception of two pairs of isolates (code numbers C1 and C15 and numbers A7 and A9), all fungal isolates were distributed in different clones. Interestingly, both *A. fumigatus* human isolates (code numbers A7 and A9), which belonged to the same cluster, exhibited resistances to all tested antifungal drugs (Fig. 3). According to the distribution of the isolates, a high degree of clonal diversity was observed, reflecting weak clonality and host specificity. The host specificity cannot be used to distribute these fungal isolates, as all animal isolates were found in different clones and there were no associations among animal or human isolates.

**Identification of ESBL- and AmpC $\beta$-lactamase-producing *E. coli* and *K. pneumoniae* isolates.** According to the modified double-disc synergy test (MDDST), 50 and 62.5%, respectively, of *E. coli* and *K. pneumoniae* isolates were ESBL producers (Fig. 4). All these isolates showed MDR patterns; meanwhile, 75 and 83.3%, respectively, of MDR *E. coli* and *K. pneumoniae* isolates were ESBL producers. Moreover, the overall occurrence of AmpC $\beta$-lactamases in this study was detected among 38.3 and 50% of *E. coli* and *K. pneumoniae*, respectively. Interestingly, 23.3 and 47.5% of *E. coli* and *K. pneumoniae* isolates were MDR and were ESBL and AmpC $\beta$-lactamase producers.

**Molecular detection of antibiotic resistance genes among bacterial and fungal isolates.** Using PCR assays, three ESBL and two AmpC $\beta$-lactamases plasmid-mediated genes were detected among our investigated MDR *E. coli* and *K. pneumoniae* isolates (Table 1 and Fig. 4). The patterns of phenotypic resistance to $\beta$-lactams and phenotypic detection of ESBL and AmpC $\beta$-lactamases were in parallel with those of PCR detections of ESBL and AmpC $\beta$-lactamase genes. Notably, $bla_{CTX-M-1}$ was the most prevalent ESBL gene among *E. coli* and *K. pneumoniae* isolates (33.3 and 37.5%, respectively). Additionally, 33.3 and 42.5% of *E. coli* and *K. pneumoniae* isolates harbored the *MOX* gene (Table 1). The data presented in Fig. 4 demonstrate that with the exception of six clusters, including five pairs of isolates (E11 and E36; E47 and K12; E5 and E42; E6 and E27; E35 and K40) and one group of three isolates (K27, E48, and K7), all isolates were distributed in different lineages. No *E. coli* or *K. pneumoniae* animal or human isolates could be placed together in the same cluster. Therefore, both *E. coli* and *K. pneumoniae* isolates showed high diversity and weak clonality within each host.

The confirmed phenotypic fluconazole resistance among *C. albicans* and *A. fumigatus* isolates was correlated with the overexpression (fold change > 1) in *C. albicans* *ERG11* and *A. fumigatus* *cyp51* genes. Accordingly, *ERG11* and *cyp51A* were upregulated among 73.3% (11/15) and 80% (8/10) of *C. albicans* and *A. fumigatus* isolates, respectively (Fig. 5). The increase in the expression levels of the investigated resistance genes ranged from 2.6- to 9.5-fold.

**Antibacterial and antifungal activities of the anthelmintic drug rafoxanide.** Fortunately, it was found that all recovered bacterial and fungal isolates were strongly inhibited by rafoxanide, with MICs ranging from 2 to 128 $\mu$g/mL (Tables S1 and S2). The growth curves of representative *E. coli* (E18), *K. pneumoniae* (K38), *C. albicans* (C3), and *A.*

**FIG 3** Legend (Continued)

to each antifungal drug (red and blue, respectively) were used to construct the heat map using the R environment program (v. 3.6.2). The code numbers on the right refer to *C. albicans* (C) and *A. fumigatus* (A) isolates from animal (C1 to C5 and A1 to A4) and human (C6 to C15 and A5 to A10) sources. With the exception of two pairs of isolates (isolates C1 and C15 and isolates A7 and A9), all fungal isolates were distributed in different clones with a high degree of antifungal diversity, reflecting weak clonality and host specificity. This heat map enables the visualization of the distribution of antimicrobial resistance among our fungal isolates. Resistance to antifungal drugs was predominant, as shown by red squares, in contrast to the sensitivity patterns, shown by blue squares. The hierarchical clustering groups similar isolates into a set of clusters, where each cluster is distinct from the others and the isolates within each cluster are similar to each other. FLZ, fluconazole; ITZ, itraconazole; Ciclo, ciclopirox; AP, amphotericin B; NY, nystatin; CASP, caspofungin; TERB, terbinafine; 5FC, 5-fluorocytosine.

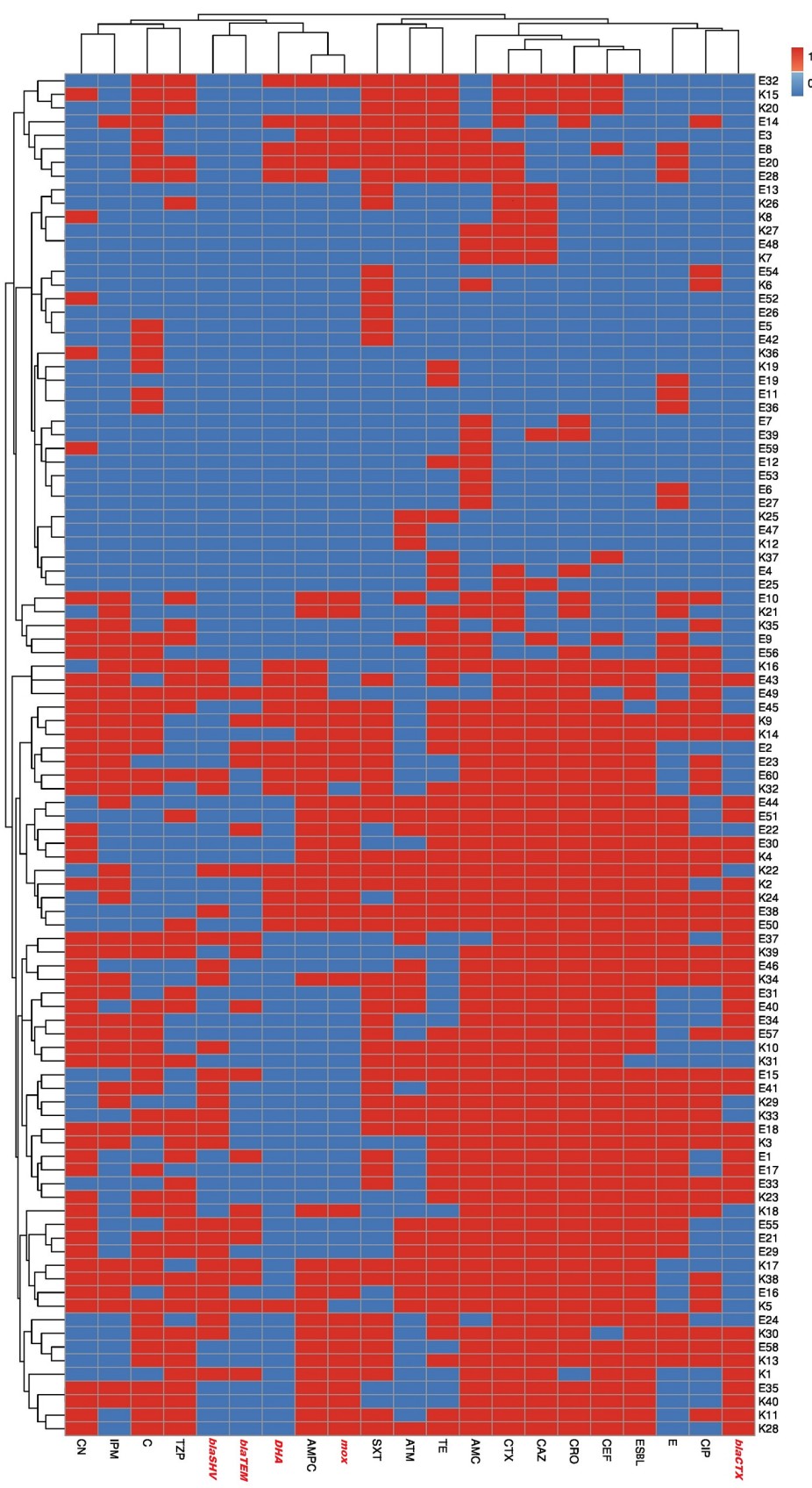

**FIG 4** Heat map showing hierarchical clustering of the investigated *E. coli* and *K. pneumoniae* isolates based on the occurrence of antimicrobial resistance, ESBL and AmpC types, and resistance genes ($bla_{CTX-M-1}$, $bla_{TEM-1}$,

**TABLE 1** Frequency of plasmid-mediated ESBL and AmpC $\beta$-lactamase genes among *E. coli* and *K. pneumoniae* isolates

| Resistance gene(s) | No. (%) of isolates harboring gene(s) | |
|---|---|---|
| | *E. coli* (n = 60) | *K. pneumoniae* (n = 40) |
| $bla_{CTX-M-1}$ | 20 (33.3) | 15 (37.5) |
| $bla_{SHV}$ | 14 (23.3) | 13 (32.5) |
| $bla_{TEM}$ | 10 (16.7) | 8 (20) |
| $bla_{TEM}$, $bla_{SHV}$ | 5 (8.3) | 1 (2.5) |
| $bla_{TEM}$, $bla_{CTX-M-1}$ | 2 (3.3) | 2 (5) |
| $bla_{SHV}$, $bla_{CTX-M-1}$ | 3 (5) | 4 (10) |
| $bla_{TEM}$, $bla_{SHV}$, $bla_{CTX-M-1}$ | 2 (3.3) | 1 (2.5) |
| *MOX* | 20 (33.3) | 17 (42.5) |
| *DHA* | 13 (21.7) | 7 (17.5) |
| *MOX* and *DHA* | 10 (16.7) | 4 (10) |

*fumigatus* (A9) isolates which showed the highest resistance patterns and were cultured in Mueller-Hinton broth in the absence or presence of 1/4×, 1/2×, and 3/4× MIC of rafoxanide are presented as optical density at 600 nm ($OD_{600}$) (Fig. S1) and CFU per milliliter (Fig. S2).

***In vitro* activity of rafoxanide in combination with antimicrobials against bacterial and fungal isolates.** Owing to the higher frequencies of both *E. coli* and *K. pneumoniae* isolates resistant to cefotaxime, the activity of cefotaxime was tested in combination with rafoxanide. The MICs of cefotaxime were reduced when the drug was combined with sub-MIC levels of rafoxanide against all tested *E. coli* and *K. pneumoniae* isolates (Fig. 6). Treatment with sub-MICs of rafoxanide resulted in 16.7% (10/60) of *E. coli* and 22.5% (9/40) of *K. pneumoniae* isolates shifting from cefotaxime resistance (MICs $\geq$ 4 $\mu$g/mL) to cefotaxime susceptibility (MICs $\leq$ 1 $\mu$g/mL). A checkerboard assay revealed remarkable synergistic activities between rafoxanide and cefotaxime against all *E. coli* and *K. pneumoniae* isolates, with fractional inhibitory concentration index (FICI) values of $\leq$0.5 (Table S2).

Fortunately, exposure of fungal isolates to sub-MIC rafoxanide concentrations in combination with fluconazole led to a shift in the resistance patterns of 63.2% (12/19) of *C. albicans* and *A. fumigatus* isolates with MICs below the resistance breakpoint (Table S1). Additionally, there was a reduction in the MICs of fluconazole for the rest of the resistant isolates (7/19 [36.8%]) when they were treated with sub-MICs of rafoxanide, but these values remained above the resistance breakpoint (Fig. 5 and Table S1).

**Resistance gene expression after exposure of bacterial and fungal isolates to rafoxanide.** The entire sets of bacterial and fungal resistance genes were found to be markedly downregulated after exposure of the examined isolates to sub-MICs of rafoxanide. Downregulation of the tested resistance genes were recorded for all treated *E. coli* and *K. pneumoniae* isolates with rafoxanide (fold change < 1). Of note, there was no significance difference between the mean fold changes of the tested resistance genes among both *E. coli* and *K. pneumoniae* isolates (P value = 0.9) (Fig. S3A). Most noticeably, the real-time reverse transcription-PCR (rRT-PCR) results indicated that $bla_{CTX-M-1}$ was the most downregulated gene, in contrast to *DHA*, among the investigated isolates (Fig. 6 and Table S3). Rafoxanide reduced the transcription levels of $bla_{CTX-M-1}$ among *E. coli* and *K. pneumoniae* isolates, with fold changes (means $\pm$ standard deviations [SD]) of 0.286 $\pm$ 0.070 and

**FIG 4** Legend (Continued)

$bla_{SHV}$, *MOX*, and *DHA*) using the R environment program (v. 3.6.2). In the heat map, red and blue indicate resistance and sensitivity to a particular antimicrobial agent and to the presence and absence of certain beta-lactamase types and resistance genes, respectively. The code numbers on the right refer to *E. coli* (E) and *K. pneumoniae* (K) isolates from animal (E1 to E20 and K1 to K12) and human (E21 to E60 and K13 to K40) sources. With the exception of six clusters, including five pairs of isolates (E11 and E36; E47 and K12; E5 and E42; E6 and E27; and E35 and K40) and one group of three isolates (K27, E48, and K7), all isolates were distributed in different lineages, reflecting high diversity and weak clonality within each host. AMC, amoxicillin-clavulanic acid; CTX, cefotaxime; CAZ, ceftazidime; CRO, ceftriaxone; CEF, cefepime; CIP, ciprofloxacin; ATM, aztreonam; C, chloramphenicol; E, erythromycin; CN, gentamicin; SXT, trimethoprim-sulfamethoxazole; TZP, piperacillin-tazobactam; IPM, imipenem; TE, tetracycline.

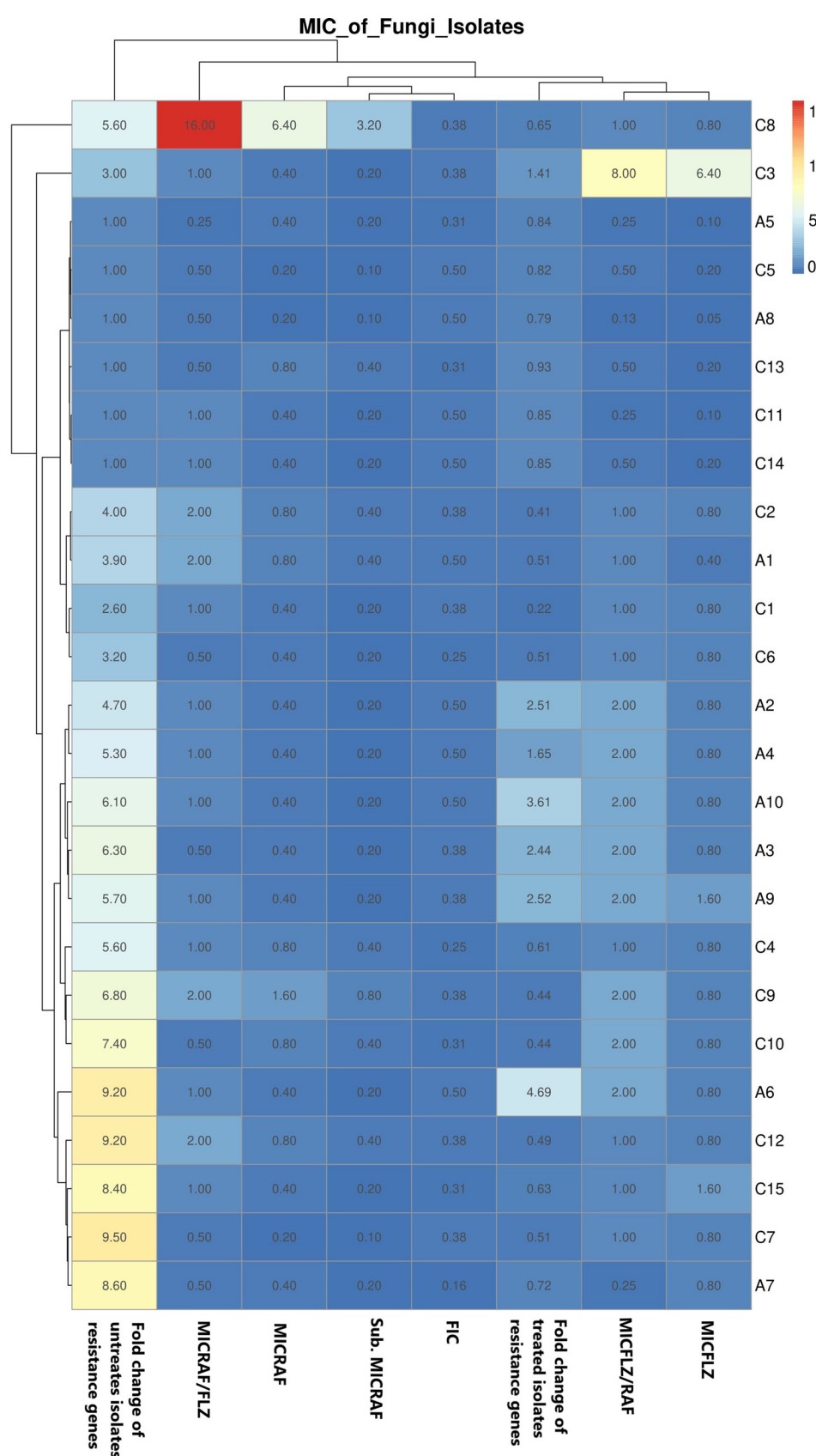

**FIG 5** Heat map and hierarchical clustering of the investigated *C. albicans* and *A. fumigatus* isolates based on the *in vitro* activity of rafoxanide alone and in combination with fluconazole. Red indicates upregulation of *C. albicans*

$0.272 \pm 0.079$ and fold change values ranging from 0.128 to 0.392 and 0.125 to 0.392, respectively. The $bla_{TEM-1}$, $bla_{SHV}$, and *MOX* genes were downregulated, but to a lesser extent, after exposure to rafoxanide (Fig. 6).

Notably, the expression levels of *C. albicans ERG11* and *A. fumigatus cyp51A* resistance genes were detected in the untreated and treated fungal isolates with rafoxanide. Interestingly, upregulation of these resistance genes was observed for all fungal isolates (fold change $> 1$). Meanwhile, both genes were downregulated upon exposure of the examined isolates to sub-MICs of rafoxanide, with documented significance difference between the mean fold changes of the treated and untreated isolates ($P = 0.041$) (Fig. S3B, Fig. 5, and Table S4). The changes in levels of the tested resistance genes in 48% of treated isolates ranged between 0.717-fold and 0.218-fold. Therefore, these isolates lost their resistance to fluconazole.

***In silico* modeling results.** Based on the obtained molecular docking results, bacterial citrate anion (CitA) formed hydrophobic interactions with amino acid residues M79 and Y56 and hydrogen bonds (HB) with Y56, T58, R66, H69, E80, G100, S101, L102, K109 and S124 (Fig. 7). Due to the significant differences between physicochemical properties of rafoxanide and bacterial CitA's substrate, rafoxanide could not fit into the binding site of CitA (region B in Fig. 7). Our results also showed the potential of rafoxanide to bind in the proximity of citrate's anion binding site with a very low docking score (approximately $-10$). However, rafoxanide could potentially affect the homodimerization of the protein via binding to the molecular surface of CitA's homodimer structure, close to the binding site between two subunits of homodimer structure (region A in Fig. 7), with a docking score of $-24.36$. Rafoxanide formed several hydrophobic interactions with CitA protein amino acid residues M21, S24, A25, V30, E31, Q34, F51, S110, P111, I112, Q113, and D114 and HB with amino acid P27 (Fig. 7). Based on the rescoring function, the obtained docking score of $-24.36$ is approximately equal to $-8.59$ kcal/mol.

Additionally, analysis of the established molecular docking studies conducted to investigate modes of rafoxanide binding at the fungal sensor histidine kinase revealed that rafoxanide could interact with fungal histidine kinase with a binding score of $-5.88$ kcal/mol, with a root mean square deviation (RMSD) of 1.68 Å. Notably, the oxygen of the hydroxyl group at position 2 of the benzamide moiety of rafoxanide could form an H bond with Gln40 at a distance of 3.35 Å (Fig. 8).

**Results of the mouse survival assay.** The results of the mouse survival assay were in accordance with those of the checkerboard assay, *in silico* modeling, and resistance gene expression analysis after exposure of the examined isolates to the sub-MICs of rafoxanide. There were significant differences in the cumulative protection levels among the negative-control group and the challenged groups treated with cefotaxime (20 mg/kg) or fluconazole (10 mg/kg) either alone or with rafoxanide ($P < 0.05$). Regarding the groups that were challenged and treated with cefotaxime-rafoxanide or fluconazole-rafoxanide, different protection levels (percent survivors without symptoms) were detected (Fig. 9). The highest cumulative protection level (80 [16/20]) was recorded for the group challenged with MDR *E. coli* (E18) challenged and treated with cefotaxime-rafoxanide. Equal cumulative protection levels (70 [14/20]) were observed among groups challenged with *K. pneumoniae* (K38) and *C. albicans* (C3) and treated with cefotaxime-rafoxanide and fluconazole-rafoxanide, respectively; meanwhile, the lowest cumulative protective level (60 [12/20]) was observed among the group challenged with *A. fumigatus* (A9) and treated with fluconazole-rafoxanide (Fig. 9).

**FIG 5** Legend (Continued)
*ERG11* and *A. fumigatus cyp51A* resistance genes and high MICs and FICI values, while the darker blue color represents downregulation of resistance genes and low MICs and FICI values. MICs of rafoxanide ($MIC_{RAF}$), fluconazole ($MIC_{FLZ}$), fluconazole used with sub-MICs of rafoxanide ($MIC_{FLZ+RAF}$), and rafoxanide used in combination with fluconazole ($MIC_{RAF+FLZ}$) and sub-MICs of rafoxanide are in micrograms per milliliter, and other values represent the expression levels of the resistance genes *C. albicans ERG11* and *A. fumigatus cyp51A* in the untreated and rafoxanide-treated fungal isolates. The code numbers on the right refer to *C. albicans* (C) and *A. fumigatus* (A) isolates. $MIC_{RAF}$, $MIC_{RAF+FLZ}$, and sub-MICs of rafoxanide are divided by 10.

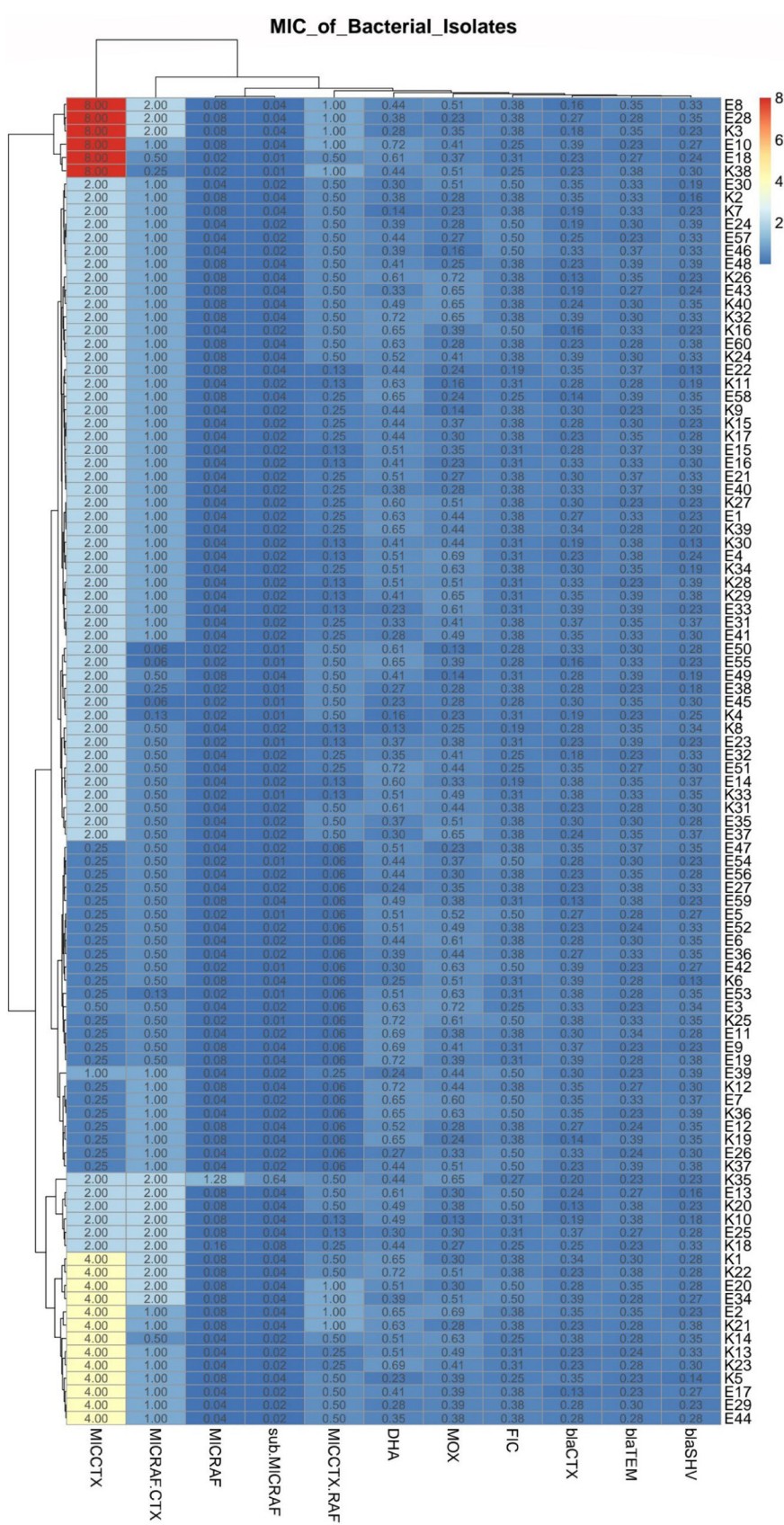

**FIG 6** Heat map and hierarchical clustering of the investigated *E. coli* and *K. pneumoniae* isolates based on the *in vitro* activity of rafoxanide alone and in combination with cefotaxime. Red indicates

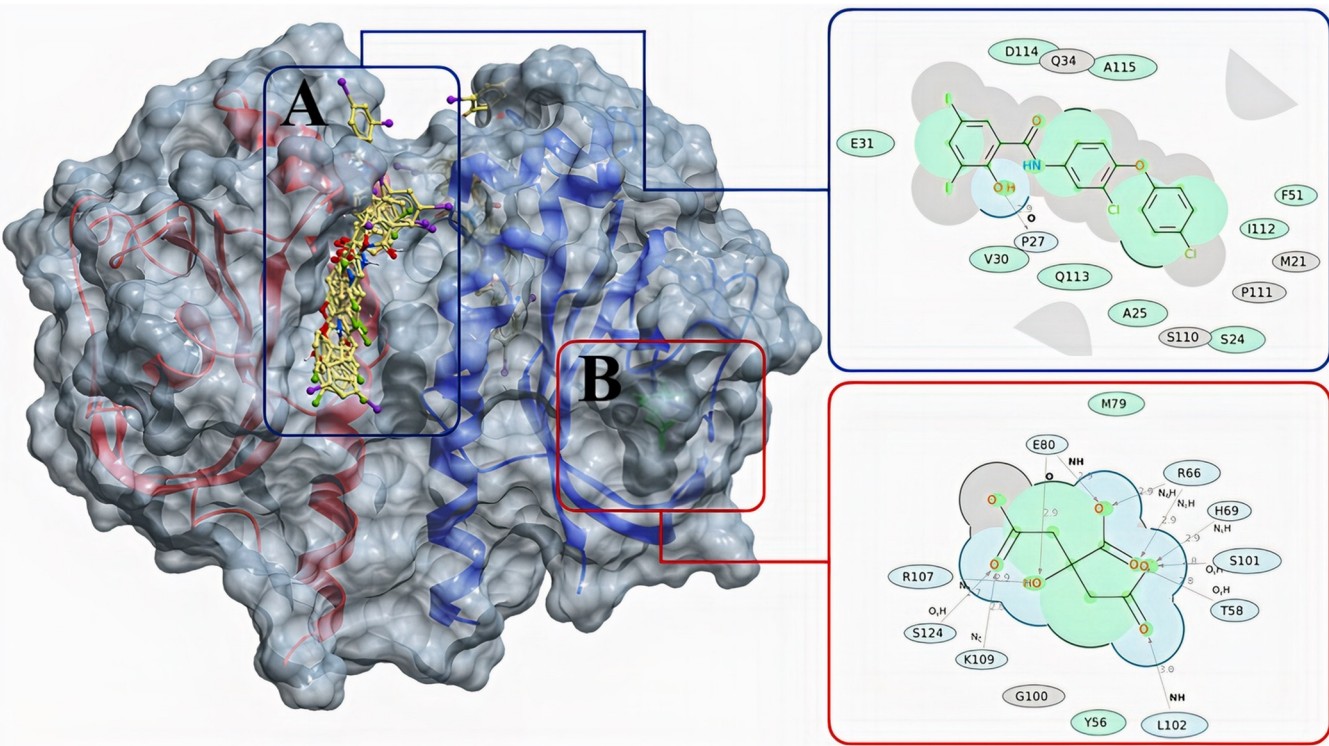

**FIG 7** Binding modes of rafoxanide (A) and citrate anion (B) and 2D interaction diagrams of corresponding ligands with bacterial CitA protein (PDB ID 2J80). Green shading in 2D interaction diagrams represents hydrophobic interactions, gray areas are accessible surfaces, and dotted arrows represent hydrogen bonds.

## DISCUSSION

The spread of antimicrobial resistance among fungal and bacterial strains is a relevant crisis for both animal and human health, leading to treatment failure. Furthermore, the evolution of resistance among *Staphylococcus aureus* and *Klebsiella* species, in addition to *C. albicans* and *A. fumigatus*, has further compounded this problem (22, 23). We found that $bla_{CTX-M}$-producing strains represented the dominant genotype among our bacterial isolates, consistent with a previous study (24). Regarding fungal infections, a high prevalence of azole-resistant strains has been observed among infected individuals (25). In accordance with our findings, *C. albicans* and *A. fumigatus* show patterns of high resistance to most available antifungals (26). This resistance has been attributed to the upregulation of both *ERG11* and *cyp51A* among resistant *C. albicans* and *A. fumigatus* strains (27, 28), with changes ranging from 2.6- to 9.5-fold.

Advances in the discovery of the new antimicrobial agents have occurred slowly. Therefore, there is an urgent need to find new and alternative therapies (29). Recently, drug repurposing, or drug repositioning, has gained the attention of many researchers owing to the low monetary cost and the minimum risk of failure (30). In this study, we observed promising antimicrobial activities of the antiparasitic drug rafoxanide. According to *in vitro* antimicrobial assays, rafoxanide has antibacterial and antifungal activities against the investigated isolates. Similarly, antimycobacterial activity of salicylanilide derivatives such as oxyclozanide and rafoxanide has also been reported (31). Moreover, all salicylanilide derivatives have shown significant antibacterial activities

**FIG 6 Legend (Continued)**

upregulation of $bla_{CTX-M-1}$, $bla_{TEM-1}$, $bla_{SHV}$, *MOX*, and *DHA* and high MICs and FICI values, while the darker blue color represents downregulation of resistance genes and low MICs and FICI values. MICs of rafoxanide ($MIC_{RAF}$), cefotaxime ($MIC_{CTX}$), cefotaxime used with sub-MICs of rafoxanide ($MIC_{CTX+RAF}$), and rafoxanide used in combination with cefotaxime ($MIC_{RAF+CTX}$), and sub-MICs of rafoxanide are in micrograms per milliliter; other values represent the expression levels of the investigated resistance genes The code numbers on the right refer to *E. coli* (E) and *K. pneumoniae* (K) isolates. $MIC_{RAF}$ and $MIC_{RAF+CTX}$ are divided by 100.

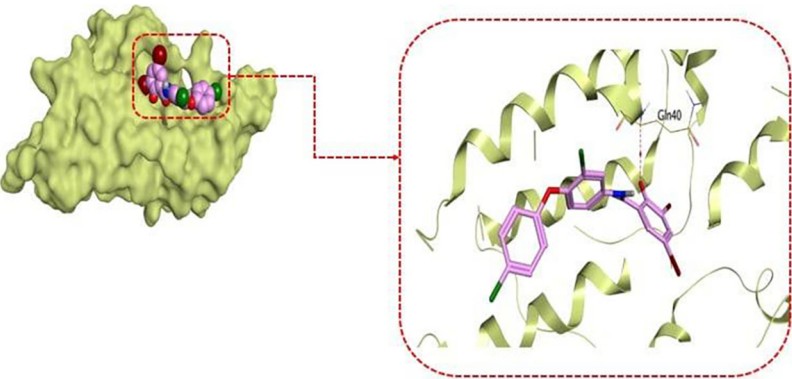

**FIG 8** 3D interaction and 3D protein positioning of rafoxanide at the sensory fungal histidine kinase with PDB ID 1OXB.

against Gram-positive strains, including methicillin-resistant *Staphylococcus aureus*, with MICs of $\geq$0.98 $\mu$mol/L (32). The MICs of rafoxanide against the tested Gram-negative bacteria in the current study ranged from 2 to 128 $\mu$g/mL, which is still below the MICs reported in previous studies (33, 34). The variation in the MICs in this study might reflect the heterogeneous nature of the tested isolates, which were isolated from different sample types, hosts, geographic areas, and genetic backgrounds.

Although several studies suggested multiple mechanisms for halogenated salicylanilide that might contribute to killing of bacteria (35), the inhibition of HK proteins remains the most acceptable pathway for this drug. It is well documented that the nucleus of salicylanilides, especially halogenated derivatives such as rafoxanide, has a TCS-inhibitory activity through its direct action on HK proteins, leading to inhibition of

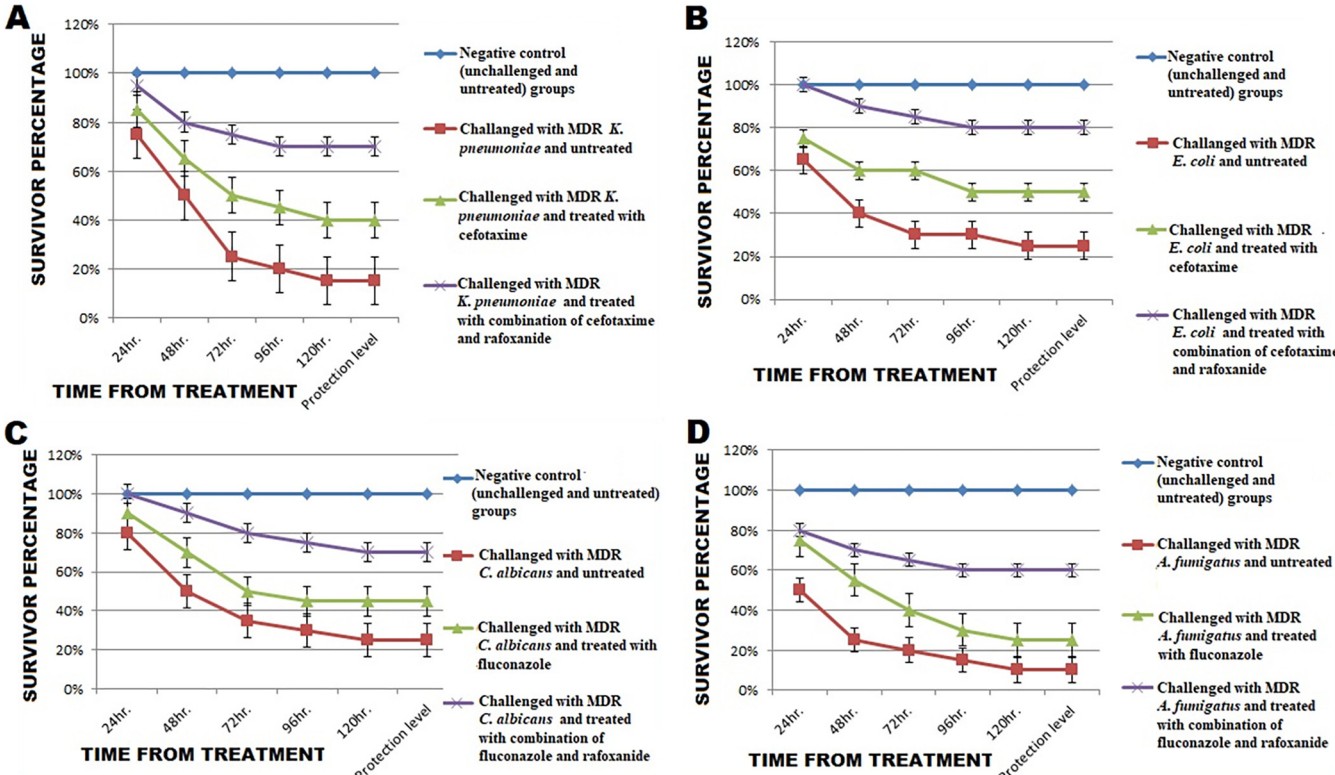

**FIG 9** Survival analysis of albino mice following intraperitoneal and intravenous injection of untreated and treated MDR bacterial (*K. pneumoniae* [A] and *E. coli* [B]) and fungal (*C. albicans* [C] and *A. fumigatus* [D]) isolates (1 $\times$ 10$^8$ CFU/mL) with sub-MIC levels of cefotaxime or fluconazole alone or in combination with rafoxanide every 24 h for 3 successive days, with standard errors. The negative-control group remained unchallenged and untreated. The survival rates (percent) of mice (20/group or subgroup) were recorded over 5 days.

bacterial growth (35, 36). The C-terminal catalytic domain of HK proteins is the direct site of binding with halogenated salicylanilides, causing protein aggregation and thus kinase inactivation (35, 36). The molecular docking in this study confirmed the binding affinity of rafoxanide with bacterial and fungal HK proteins, with binding scores of −8.59 and −5.88 kcal/mol, respectively, indicating its high efficiency in binding with the active sites of HK proteins. Based on the above-mentioned findings, we could confirm that the promising activities and the protective effects of rafoxanide can be attributed to its affinity for binding and occupying HK receptors in bacterial and fungal cells.

The higher doses of rafoxanide (MICs ranging from 2 to 128 $\mu$g/mL) required in the current study to inhibit bacterial and fungal growth cannot be used in clinical fields. Thus, this drug must be used in lower doses in combination with other antimicrobial drugs to increase the possibility of therapeutic efficiency against drug-resistant bacterial and fungal infections. In the same context, our results confirmed that most available antibacterial and antifungal drugs have lost their efficacy (37). Accordingly, resistance to cefotaxime and fluconazole drugs was the most prominent phenotype among bacterial and fungal isolates, respectively, in this study. In the light of the above, we evaluated the antimicrobial activities of cefotaxime and fluconazole when they were used with sub-MICs of rafoxanide, which could be used clinically. Fortunately, we recorded promising results for these combinations. A major shift in the antimicrobial patterns for the treated isolates with rafoxanide was observed in our *in vitro* and *in vivo* studies. This shift was attributed to the downregulatory effects of rafoxanide on the resistance genes. Notably, the accepted explanations for the role of rafoxanide (halogenated salicylanilide) in the downregulation of the resistance genes are the inhibition of HK proteins of TCSs, which allows the adaptation of bacteria and fungi through coordinated responses to environmental changes (38). For $\beta$-lactam antibiotics, the TCS is involved in the upregulation and activation of $\beta$-lactamase (39, 40). The HK proteins act as transmembrane signals, which induce the production of $\beta$-lactamase enzymes through the upregulation of $\beta$-lactamase genes (41). Signal transduction and subsequent regulation of gene expression systems were interrupted upon targeting of the HK (38, 42). Therefore, the inhibition of the TCS, especially HK proteins, downregulates the $\beta$-lactamase genes (i.e., turns the gene off) and subsequently suppresses the production of $\beta$-lactamase.

The mouse protection assay gave further evidence of the promising activities of rafoxanide when used with other antimicrobial drugs. The mouse protection assay is an *in vivo* assay used to provide insights into the antimicrobial activities of drugs (43). Finally, a limitation of the current study is its inability to determine the precise mode of action of rafoxanide and its role in overcoming microbial resistance upon treatment with the least effective antimicrobials at the molecular level with regard to specific resistance gene regulators and at the protein level other than molecular docking. Therefore, more specific research on these aspects could help in interpreting the exact mode of action of rafoxanide.

**Conclusion.** The therapeutic switching of existing drugs to reduce the evolution of antimicrobial resistance is considered a promising way to manage this issue. HK protein inhibition is a promising strategy to fight the ongoing spread of resistant pathogens. Our report confirms the therapeutic potential of rafoxanide, at a tolerable dose, when coadministered with other antifungal and antibacterial agents. Accordingly, most resistant bacterial and fungal cells become susceptible to antimicrobial therapies when the antimicrobials are combined with rafoxanide. Rafoxanide at a suitable dose had HK-antagonistic activities, which disrupted the abilities of bacterial and fungal cells to adapt to stress conditions. Finally, we hope that the results of our study will prompt the therapeutic industries to develop new drug combinations against drug-resistant pathogens.

## MATERIALS AND METHODS

**Ethics.** All samples collected from patients at Zagazig University and Port Said University were used in the antimicrobial susceptibility testing. Therefore, informed consent was required after explanation of the purpose and nature of the research. The *in vivo* mouse protection assay was done in accordance with the recommendations of the Ethical Scientific Research Committee, Faculty of Pharmacy, Port Said

University (REC.PHARM.PSU. 22-3) and following the ARRIVE guidelines, the U.K. Animals (Scientific Procedures) Act 1986, and related guidelines (ECAHZU, 23 August 2015).

**Isolation and identification of bacterial and fungal isolates.** One hundred ear swabs were obtained from patients suffering from otitis media and dogs presenting with clinical otitis (50 each) for isolation of *C. albicans* and *A. fumigatus*. Moreover, 200 stool samples were collected from diarrhetic patients and animals (100 each) for isolation of *E. coli* and *K. pneumoniae*. First, MacConkey and Levine's eosin methylene blue selective agars (Oxoid, UK) were used for bacterial isolation. The recovered isolates were identified according to standard phenotypic methods (44) and were confirmed using API-20E strips (bioMérieux, Marcy l'Étoile, France). Moreover, PCR assays targeting *16S* rRNA and 16S-23S internal transcribed spacer genes were carried out to confirm *E. coli* (45) and *K. pneumoniae* (46) isolates, respectively. *E. coli* ATCC 25922, *K. pneumoniae* ATCC 13883, and *Staphylococcus aureus* ATCC 6538, which were purchased from Animal Production Research Institute, Giza, Egypt, were used as positive and negative controls, respectively. Second, Sabouraud dextrose agar was used for isolation of *C. albicans* and *A. fumigatus* isolates. The genus of fungal isolates was determined on the basis of cultural and morphological characteristics (47). Observed characteristics were recorded and compared with established identification keys (48). Moreover, the species was determined by sequencing of *18S* rRNA PCR products (49).

**Antimicrobial susceptibility testing.** The susceptibilities of the recovered bacterial and fungal isolates to commonly used antimicrobials were determined via Kirby-Bauer disc diffusion and broth microdilution methods using Mueller-Hinton agar (MHA) and commercial antimicrobial and antifungal disks (Oxoid, UK). The antimicrobial agents tested were amoxicillin-clavulanic acid (AMC), cefotaxime (CTX), ceftazidime (CAZ), ceftriaxone (CRO), cefepime (CEF), ciprofloxacin (CIP), aztreonam (ATM), chloramphenicol (C), erythromycin (E), gentamicin (CN), trimethoprim-sulfamethoxazole (SXT), piperacillin-tazobactam (TZP), imipenem (IPM), and tetracycline (TE). The antifungal agents tested were fluconazole (FLZ), itraconazole (ITZ), ciclopirox (Ciclo), amphotericin B (AP), nystatin (NY), caspofungin (CASP), terbinafine (TERB), and 5-fluorocytosine (5FC). Both techniques were performed in accordance with Clinical and Laboratory Standards Institute (CLSI) recommendations (50, 51). In brief, MHA, Sabouraud dextrose agar, and MHA with 2% glucose and 0.5 mg/L methylene blue dye (which enhances the zone edge definition) were swabbed with bacterial, filamentous fungal (*A. fumigatus*), and yeast (*C. albicans*) solutions, respectively (52). The suspensions of all isolates were adjusted spectrophotometrically to optical densities equivalent to 0.5 McFarland standard solutions. The entire agar surfaces were then streaked with sterile cotton swabs dipped in the adjusted suspensions three times. After the plates dried, antimicrobial discs were gently pressed into the agar, and the plates were incubated at 37°C for 24 h for bacterial isolates and at 35°C for 24 h to 5 days for fungal isolates. The isolates were classified as sensitive or resistant to the antimicrobial drugs based on the diameters of the inhibition zones according to the interpretative criteria developed by CLSI. Furthermore, the broth microdilution method was used to confirm the disc diffusion results. This involved the preparation of 2-fold dilutions of tested antimicrobial drugs using Mueller-Hinton broth (Oxoid, UK) for bacterial isolates and Roswell Park Memorial Institute 1640 medium (Sigma, Germany) with L-glutamine but without sodium bicarbonate and supplemented with 2% glucose for fungal isolates (53, 54) in 96-well microtiter plates containing the prepared microbial suspensions. Positive and negative controls, consisting of microbial inocula without antimicrobial drugs and antimicrobial drugs without microbial inocula, respectively, were included. The microtiter plates were incubated at 37°C for 24 h for bacterial isolates and at 35°C for 48 h for fungal isolates, and then the MICs were determined and interpreted (55, 56).

Isolates showing resistance to one drug each in three or more different classes of antimicrobial agents were identified as MDR (2).

**Identification of ESBL-producing *E. coli* and *K. pneumoniae* isolates.** An MDDST with four cephalosporins—three third generation (cefotaxime, ceftriaxone, and ceftazidime) and one fourth generation (cefepime)—in addition to amoxicillin-clavulanic acid was used to identify the ESBL-producing bacterial isolates (57). The discs of third- and fourth-generation cephalosporins were placed 15 and 20 mm (center to center) from the amoxicillin-clavulanate disc, respectively. Any distortion or increase in the zone toward the disc of amoxicillin-clavulanic acid was considered a positive result for ESBL production.

**Detection of AmpC β-lactamase production using the AmpC disc test.** A lawn culture with the cefoxitin-susceptible strain *E. coli* ATCC 25922 was inoculated on a Mueller-Hinton agar plate. Several colonies of *E. coli* and *K. pneumoniae* were inoculated on the AmpC disks (sterile filter paper disks containing Tris-EDTA). A cefoxitin disc was placed next to the AmpC disc (almost touching) on the inoculated plate. The plates were incubated overnight at 37°C. A flattening or indentation of the cefoxitin inhibition zone in the vicinity of the disc was considered a positive result (58).

**Molecular detection of antibiotic resistance among bacterial and fungal isolates.** Extraction of DNA from bacterial isolates grown overnight in brain heart infusion broth was carried out using a QIAprep Spin miniprep kit (Qiagen, Germany) following the manufacturer's instructions. All PCR assays were conducted in triplicate, in total reaction volumes of 25 μL containing 12.5 μL of Emerald Amp GT PCR master mix (TaKaRa, Japan), 1 μL of each primer (20 pmol), 3 μL of each template DNA, and 7.5 μL of PCR-grade water. PCR amplifications of ESBL and AmpC β-lactamase genes were done as previously described (59–61). The PCR-amplified products were analyzed using agarose gel electrophoresis under a UV transilluminator (Spectroline, USA).

For molecular detection of fluconazole resistance among fungal isolates, analysis of expression of *C. albicans* ERG11 and *A. fumigatus* cyp51A genes was carried out using an rRT-PCR assay. Total RNA was isolated from the fungal isolates grown overnight in Sabouraud dextrose broth medium using the RNeasy minikit (Qiagen, Germany) according to the manufacturer's instructions. The rRT-PCR assays were carried out, in triplicate, in an MX3005P real-time PCR apparatus (Stratagene Co., USA) using a

**TABLE 2** Primer sequences, numbers of cycles, and annealing temperatures for PCR amplifications of resistance genes among bacterial and fungal isolates

| | Primer direction[a] | Primer sequence (5′–3′) | No. of cycles | Annealing temp (°C) | Reference |
|---|---|---|---|---|---|
| $bla_{CTX-M-1}$ | F | CCCATGGTTAAAAAACACTGC | 30 | 57 | 77 |
| | R | CAGCGCTTTTGCCGTCTAAG | | | |
| $bla_{TEM}$ | F | ATAAAATTCTTGAAGACGAAA | 30 | 53 | 59 |
| | R | GACAGTTACCAATGCTTAATC | | | |
| $bla_{SHV}$ | F | ATTTGTCGCTTCTTTACTCGC | 30 | 52 | 60 |
| | R | TTTATGGCGTTACCTTTGACC | | | |
| DHA | F | AACTTTCACAGGTGTGCTGGGT | 25 | 64 | 61 |
| | R | CCGTACGCATACTGGCTTTGC | | | |
| MOX | F | GCTGCTCAAGGAGCACAGGAT | 25 | 64 | 61 |
| | R | CACATTGACATAGGTGTGGTGC | | | |
| ERG11 | F | TGGAGACGTGATGCTG | 40–50 | 62 | 62 |
| | R | AGTATGTTGACCACCCATAA | | | |
| cyp51A | F | TCATTGGGTCCCATTTCT | 45 | 61 | 63 |
| | R | GCACGCAAAGAAGAACTTG | | | |

[a]F, forward; R, reverse.

QuantiTect SYBR green RT-PCR kit (Qiagen, Germany) and specific primers targeting resistance genes, as detailed elsewhere (62, 63). The constitutive expression levels of *18S* rRNA housekeeping genes were used to normalize the expression of the genes of interest. A complete list of the primers targeting resistance genes as well as the numbers of cycles and annealing temperatures used in the current study are provided in Table 2.

**Rafoxanide.** Rafoxanide powder [3′-chloro-4′-(4-chlorophenoxy)-3,5-diiodosalicylanilide] with the chemical formula of $C_{19}H_{11}Cl_2I_2NO_3$ and a molecular weight of 626.01 g/mol was purchased from Sigma-Aldrich (St. Louis, MO).

**Antibacterial and antifungal activities of the anthelmintic drug rafoxanide alone and in combination with other antimicrobials.** Antibacterial and antifungal activities of rafoxanide alone were affirmed by assessing its MICs against the examined isolates via the broth microdilution technique. To evaluate the *in vitro* combinations between the least effective antimicrobials against the tested bacterial and fungal isolates and rafoxanide, a checkerboard method was carried out, in triplicate, to determine the FICI values. The interpretation of the results obtained was as follows: antagonism, FICI > 4; synergy, FICI ≤ 0.5; indifference, 1 < FICI ≤ 4; and additive, 0.5 < FICI ≤ 1 (64, 65).

**Growth curve analysis.** Representative bacterial and fungal isolates inoculated in Mueller-Hinton broth were individually treated with 1/4×, 1/2×, and 3/4× MIC rafoxanide. Bacterial and fungal cultures without rafoxanide served as controls. Later, bacterial and fungal growth was measured over a period of 24 h by determining the $OD_{600}$ and the viable cell counts (CFU), and growth curve graphs of $OD_{600}$ and counts (in CFU per milliliter) versus time (in hours) were then plotted. To ensure the reproducibility of the results, all growth experiments were carried out in three biological replicates.

**Resistance gene expression analysis for bacterial and fungal isolates treated with rafoxanide.** rRT-PCR analysis was used to measure the transcript levels of the resistance genes in *E. coli* and *K. pneumoniae* encoding ESBLs ($bla_{CTX-M-1}$, $bla_{TEM-1}$, and $bla_{SHV}$) and AmpC β-lactamase (*MOX* and *DHA*) and of the *C. albicans ERG11* and *A. fumigatus cyp51A* genes after treatment of the isolates with sub-MICs of rafoxanide, as detailed above, using *16S* rRNA, 16S-23S internal transcribed spacer, and *18S* rRNA housekeeping genes to normalize the expressions of the target genes in *E. coli* and *K. pneumoniae* and in fungal isolates, respectively. A melting curve analysis was applied, postamplification, to confirm the amplicon specificity. The relative gene expression was calculated using the comparative cycle threshold ($C_T$) ($2^{-\Delta\Delta CT}$) method (66), and the results were expressed as fold changes.

***In silico* computer-based modeling technologies.** Molecular docking studies were employed examine the potential of rafoxanide to inhibit bacterial and fungal histidine kinases. The crystal structure of the citrate-bound periplasmic domain of bacterial sensor histidine kinase CitA (Protein Data Bank [PDB] ID 2J80; RMSD = 1.60 Å) was downloaded from the PDB (67) and used for the molecular docking studies. Moreover, the X-ray structure of sensory fungal histidine kinase protein was investigated and downloaded from the PDB (ID 1OXB; RMSD = 1.68 Å) (68). Consequently, the target protein chain was determined and then protonated. Thereafter, the broken bonds of the target protein were corrected, followed by energy minimization to be ready for the docking process as previously discussed in detail (69). The rafoxanide structure determined here was imported into a database to be saved as an MDB file to be ready for the docking process. Visualization of docking results was done via three-dimensional (3D) representation of docked complex and 2D interaction diagrams of cocrystallized ligand and rafoxanide. The regular score, which is augmented with a directional hydrogen bonding term, was used for evaluating the interaction energy of studied compounds and protein (70). Docking effort was set to 10.

**Mouse survival assay.** We further evaluated the efficacy of rafoxanide-antimicrobial combinations using mouse survival assay as described elsewhere (34, 71), with some modifications. Two hundred sixty male albino mice with a weight range of 16 to 18 g were divided into five groups; one unchallenged and untreated (20 mice) serving as a negative-control group and four equal challenged groups (60 mice each). Mice in the first two challenged groups were intraperitoneally (i.p.) injected with 0.1 mL of MDR

multivirulent bacterial (*E. coli* and *K. pneumoniae*) solutions containing $1 \times 10^8$ CFU/mL with total CFU equaling $10^7$ (71). For fungal isolates, the mice in the other two challenged groups were immunosuppressed by i.p. injection of 200 mg/kg cyclophosphamide for 3 successive days (72). After that, the immunosuppressed mice were injected intravenously with 0.1-mL solutions containing $2.5 \times 10^5$ blastospores and $0.8 \times 10^6$ conidia of MDR multivirulent *C. albicans* (73) and *A. fumigatus* (74), respectively. Four hours after bacterial and fungal inoculation, each group was subdivided into three subgroups; one that was challenged only, one that was challenged and treated with the least effective antimicrobials against the tested isolates at concentrations equal to twice their MICs, and one that was challenged and treated intravenously with sub-MICs of rafoxanide (34) in combination with the same antimicrobials every 24 h for 3 successive days (72 h). Finally, the numbers of surviving mice were recorded over five successive days.

**Statistical analysis.** All statistical analyses were done based on the R packages corrplot, heat maply, hmisc, and ggpubr and GraphPad Prism (version 6; GraphPad Software Inc., USA). Moreover, all heat maps and dendrograms showing hierarchical clustering were constructed using the R environment (v. 3.6.2) (75, 76). Means and SD were used to report each experimental value. Statistical significance was determined by Student's *t* test, and *P* values of <0.05 were considered statistically significant.

**Data availability.** All data generated or analyzed during this study are included in the published article.

## SUPPLEMENTAL MATERIAL

Supplemental material is available online only.
**SUPPLEMENTAL FILE 1**, XLSX file, 0.01 MB.
**SUPPLEMENTAL FILE 2**, XLSX file, 0.01 MB.
**SUPPLEMENTAL FILE 3**, XLSX file, 0.01 MB.
**SUPPLEMENTAL FILE 4**, XLSX file, 0.01 MB.
**SUPPLEMENTAL FILE 5**, PDF file, 0.8 MB.

## ACKNOWLEDGMENTS

M.M.B. and M.I.A.-H. designed the work. A.R.E., A.I.A., and N.A.S. carried out the antimicrobial sensitivity test. The molecular techniques and data analysis were conducted by A.E., M.M.G., N.H., W.A.A., H.F.A.-H., and R.A.M. Both the bioinformatics and statistical analyses of the data were performed by M.A., A.S.A., and M.M.B. Additionally, M.M.B. and M.I.A.-H. wrote the initial draft of the paper. All authors have read and agreed to the published version of the manuscript.

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
