## [Reviewer comments · Microbiology Spectrum]

Microbiology Spectrum

Therapeutic switching of rifaxanide: A new approach to fight resistant bacteria and fungi

Mahmoud Bendary, marwa Abd El-Hamid, Amira Abousaty, Arwa Elmanakhly, wala Alshareef, Rasha Mosbah, Majid Alhomrani, Mohammed ghoneim, Amr Elkelish, Nada Hashim, Abdulhakeem Alamri, Helal Al-Harhi, and nasreen safwat

Corresponding Author(s): Nada Hashim, University of Gezira Faculty of Medicine

Review Timeline:

Submission Date:	July 12, 2022
Editorial Decision:	November 14, 2022
Revision Received:	January 19, 2023
Editorial Decision:	May 9, 2023
Revision Received:	May 25, 2023
Accepted:	June 5, 2023

Editor: Francisco Uzal

Reviewer(s): Disclosure of reviewer identity is with reference to reviewer comments included in decision letter(s). The following individuals involved in review of your submission have agreed to reveal their identity: Kaan Çeylan (Reviewer #2)

Transaction Report:

DOI: <https://doi.org/10.1128/spectrum.02679-22>

November 14, 2022

Prof. Mahmoud Bendary
portsaid university
zagazig, sharquia
sharquia
Egypt

Re: Spectrum02679-22 (Therapeutic switching of rafoxanide: A new approach to fight resistant bacteria and fungi)

Dear Prof. Mahmoud Bendary:

Link Not Available

Sincerely,

Francisco Uzal

Journals Department
Reviewer comments:

Reviewer #1 (Comments for the Author):

Please see attached document.

Staff Comments:

Preparing Revision Guidelines

Please return the manuscript within 60 days; if you cannot complete the modification within this time period, please contact me. If you do not wish to modify the manuscript and prefer to submit it to another journal, please notify me of your decision immediately so that the manuscript may be formally withdrawn from consideration by Microbiology Spectrum.

The manuscript (Spectrum02679-22) '**Therapeutic switching of rafoxanide: A new approach to fight resistant bacteria and fungi**' by Bendary and Abdel-Hamid *et al.* investigates the use of the veterinary medicine, rafoxanide against a collection of bacterial and fungal isolates and examines the use of rafoxanide to re-sensitise MDR bacteria and fungi to various β -lactam antibiotics or antifungals respectively. In addition to standard antimicrobial sensitivity assays (e.g., MICs, chequerboards), Bendary and Abdel-Hamid *et al.* provide preliminary data on a possible mechanism for how rafoxanide acts to reverse antimicrobial resistance in these isolates. The approach focused almost entirely on the expression of bacterial β -lactamase genes and two critical genes, ERG11 and *cyp51A*, in the fungal species, examined. While it is clear that new strategies are required to help combat the risk posed by antimicrobial resistance, further information is necessary to support the hypothesis that rafoxanide overcomes antimicrobial resistance by inhibiting histidine kinase activity in the pathogens examined in this study. I found the manuscript difficult to read.

General Comments

1. The abstract would benefit from further clarification. For example, on lines 37-39, what does 'resistance patterns' mean? It is not apparent from reading the abstract in isolation what this term refers to, likewise for 'associated percentage'
 - a. Abstract Line 38; please consider clarifying where these isolates originated (i.e., animal, and human clinical samples, environmental samples etc.).
 - b. It is unclear in the abstract how the authors moved from examining the expression of AMR-related genes (i.e., beta-lactamases) to focusing on TCSs/histidine kinases. The author's reasoning for doing so could be clarified if the rationale for selecting and focusing on rafoxanide is explained when the compound is introduced. Regarding this, more information should be provided in the introduction establishing the role of rafoxanide as an inhibitor of TCSs.
 - c. Please edit Lines 46-47 in the abstract as there is no direct evidence provided in the manuscript to indicate that inhibition of histidine kinases protein activity occurs with rafoxanide or that this plays a role in the mechanism of resistance reversal. The only data provided to support this hypothesis is *in silico* modelling. Further, no data is provided in the manuscript to indicate this is the mechanism in the fungal isolates examined.
 - d. On line 42 of the abstract, please clarify the units associated with '<2'.
 - e. On line 44 of the abstract, please consider providing more detail regarding 'improvement in the activity,' i.e., provide x-fold changes or FICI values.
2. The methods section of the manuscript describing antimicrobial susceptibility testing is unclear. For example, what media was used for the bacterial testing, presumably cation-adjusted Mueller Hinton broth? Please consider adding more detail. Similar comments for testing of the Fungi, were the CLSI guidelines followed?
 - a. Further, how were the results (i.e., breakpoints) interpreted, i.e., according to CLSI guidelines or EUCAST guidelines?
 - b. Consider combining the susceptibility testing method section with the chequerboard section, as there is some overlap in the text.
3. In the applicable Figure legends, please describe how the heatmap and hierarchical clustering were performed and what software packages (with citations) were utilised.

4. Please consider describing the gene function of *MOX* and *DHA* in the introduction concerning antibiotic resistance and provide citations. More information is required about these two genes and their roles (i.e., AmpC). Line 76 would be a good place for this brief clarification.
5. Can you please provide an accurate reference for the role of rafoxanide as a TCS agonist?

Is there direct evidence for this, or is this rationale based on data with other salicylanilides? This could be made very clear in the introduction to support the role of rafoxanide as a TCS agonist.

6. Is reference 24 correctly cited? Does this citation indeed provide direct evidence that rafoxanide is a TCS agonist; I can't seem to find this information in the cited paper. Please double-check all referencing is appropriate for statements made in the manuscript.
7. Please consider editing the statement on lines 110-111. Do these citations provide direct evidence that rafoxanide alters the expression of beta-lactamase genes? Please double-check all referencing is appropriate.
8. Please consider splitting Figure 8 into three separate graphs (A, B, and C) for each microbe examined and please add the statistical analysis to the graphs in Figure 8. I have made other comments on this figure below.

Specific Comments

1. Figure S1. Can the authors provide bacterial viability data (i.e., CFU/mL?). Further, please provide a detailed figure legend and information about biological replicates.
2. Based on previously established literature, rafoxanide has been proposed to inhibit oxidative phosphorylation. Is there a reason this was not investigated further as a central mechanism behind rafoxanide-mediated resistance reversal? Please consider clarifying this in the manuscript. Doing so might help the reader understand your focused approach.
 - a. Further, other related anthelmintic drugs (such as oxyclozanide) are known to disrupt/alter the permeability of the Gram-negative cell wall. Again, the authors could provide a rationale for their narrow focus on beta-lactamase enzymes, while the literature suggests the authors could have investigated other hypotheses.
3. Given the author's hypothesis that rafoxanide inhibits histidine kinases, why were no studies conducted to examine this effect directly? This gap could be addressed in the discussion and included in future work.
4. On lines 181-184, please provide detailed information about what bacterial and fungal isolates were selected for this analysis.
 - a. Does the $\frac{1}{4}$ MIC value etc., refer to $\frac{1}{4}$ of the mean MIC or the measured MIC for the specific strain investigated?

5. The figure legend for Figure 1 is entirely unclear and requires more detail so the reader can complete a stand-alone analysis. For example, what exactly does the 'MDR pattern' refer to?
 - a. Do the 60+40 isolates refer to 100 in total (for *E. coli*)? Or does it mean 40/60 (66%) isolates had an 'MDR pattern'? When examining the figure, it's not apparent; the authors can resolve this with further description in the figure legend.
 - b. Is this 'MDR pattern' based on antimicrobial susceptibility data or the presence of resistance genes as determined by PCR etc.? I realise this is described on lines 370-371, but it should be clarified in the Figure legend text.
 - c. Further, the authors frequently use full names, e.g., *Aspergillus fumigatus* or *A. fumigatus*. Consistency is required throughout the manuscript.
 - d. Fix formatting of *E-coli* to *E. coli*.
 - e. The associated results section for Figure 1 should highlight for the reader how many isolates were of animal or human origin and if this had any impact on the susceptibility profiles.

6. I have similar comments regarding the legend of Figure 2 – more detail is required to allow stand-alone analysis.
 - a. Did the authors take, for example, all 40 *E. coli* MDR pattern isolates and individually test all 40 isolates against the described antibiotics? Was this conducted by micro broth dilution, in what media, and under what conditions for bacterial and fungal isolates? Please mention biological replicates and what threshold was used to determine resistance (i.e., EUCAST or CLSI breakpoints). The authors should include mention of this in the results section.
 - b. Please fix various formatting issues in Figure 2. Italicise bacterial and fungal names, correct capitalisation and correct mixed use of full and abbreviated names.
 - c. Note that Figure 2A x-axis ends at 90%; Fig 2B ends at 100%.

7. In Figure 3 and the associated results text as it is currently written, the value of the hierarchical clustering within a heatmap (Figure 3) is unclear. Further, the authors should provide more supporting text regarding this analysis in the results section on lines 138-146.
 - a. Was there any correlation between species and where they were isolated from (human or animal) with their antimicrobial resistance profile? It appears that there are four major clusters shown; the authors could consider commenting on these clusters, what they represent, where they were isolated from and the resistance profiles.
 - b. What method was used to perform the clustering and distance matrix? These details should be made clear in the figure legend.
 - c. It is unclear why the heat map key was included when the authors only used two colours.
 - d. It is unclear what genotypic profile traits were used to generate this figure.

8. Figure 4 and the results it represents are challenging to interpret. For example, the antibiotic resistance genes could be placed together. I found using red to highlight antibiotic resistance and the presence of an antibiotic resistance gene confusing.

9. Move lines 166-170 to the antifungal activity section
 - a. MICs are generally not written with +SDs.

- b. Further, was this mean MIC value across *all* strains examined or only 'MDR pattern' isolates? Please make this clear in the results section.
 - c. Please also clearly state the MIC values against the control ATCC strains and include this information in the text.
 - d. This section would benefit enormously from including the MIC data for the strains examined in a Table in supplementary information or the main text. Likewise, the associated FICIs should be included in this table.
10. These MIC values appear to be relatively low for rafoxanide against Gram-negative bacteria. For example, Domaloan *et al.* ([10.1038/s41429-019-0186-8](https://doi.org/10.1038/s41429-019-0186-8)) demonstrated that rafoxanide had very high MICs against *E. coli* (>256 ug/mL), and a similar result was observed by Miro-Canturri *et al.* with rafoxanide MICs against *K. pneumoniae* of >256 ug/mL. The authors should consider adding some commentary around the differences in these observed sensitivities of the stand-alone activity of rafoxanide and contrast this with their data.
11. Please clarify the statement on line 166, 'measuring the expression levels'; in the context of a section describing molecular detection by PCR, it is confusing.
 - a. Did you use RT-qPCR to detect the gene expression of ERG11 and *cyp51A*? Or was some analysis of the presence of these two fungal genes also conducted by PCR only?

This section is very confusing as you are no longer talking about molecular detection but rather the gene expression between antifungal (i.e., azole) treated and untreated fungal cells. These data could be moved to another more appropriate section of the manuscript. Consider moving this to lines 211-213.
 - b. Further, providing detailed information on the level of upregulation observed with appropriate statistical analysis would be beneficial.
 - c. Interpreting these data from Figure 5 alone is very difficult; a bar graph with relative gene expression levels would be easier to analyse and allow you to show the relevant analysis.
 - d. Please provide additional discussion on the biological relevance of a 0.218-0.7-fold change in gene expression levels.
12. It is unclear to me why the authors did not investigate changes in gene expression levels to compare A) rafoxanide treated cells, B) antibiotic or antifungal treated cells, and C) rafoxanide+antifungal (or antibiotic) treated cells. Suppose you were to compare the gene expression levels under all three conditions relative to untreated controls. In that case, you could potentially observe more significant changes in the gene expression levels and have more robust data to validate your hypothesis.
 - a. Additionally, some microorganisms produce beta-lactamase enzymes in a regulated way (i.e., in the presence of the beta-lactam antibiotic). Including the beta-lactam antibiotic in a treatment group might ensure that the beta-lactamase genes are upregulated, allowing the authors to investigate the effect of rafoxanide under more appropriate experimental conditions.
 - b. Further, if the authors believe rafoxanide-mediated changes to beta-lactamase expression play a key role in overcoming beta-lactam resistance, could they consider the importance of investigating the gene expression of specific regulators of beta-lactamase expression?
 - c. Can the authors please clarify if any of the beta-lactamase enzymes investigated in the pathogens described are under the control of two-

component regulons? Some additional text in the discussion regarding this would be helpful.

13. Figure 6 is tough to interpret

- a. Firstly, it is unclear what the exact MICs are for all strains examined against cefotaxime. By using the coloured legend, only an approximation of the MIC can be interpreted by the reader. I think this MIC data is better reported in a Table with the exact values.
- b. It is unclear which breakpoint value the authors use for cefotaxime; this should be described. For example, EUCAST sets a resistance breakpoint of ≥ 2 ug/mL (for indications other than meningitis). Please add this information to the results text.
- c. The legend for Figure 6 should indicate the sub-MIC value, and if this was consistent across all bacterial strains, please clarify what was used. i.e., $\frac{1}{4}$ MIC? This should also be made clear in the accompanying results text. Again, the use of a table would solve this issue.
- d. Lines 196-200 are difficult to understand. Do the authors mean that 12/25 fungal isolates were re-sensitised to fluconazole (below MIC breakpoints) in the presence of sub-inhibitory rafoxanide? At the same time, other strains demonstrated a reduction in the MICs, but this was not clinically relevant (i.e., not below MIC breakpoints?). Please clarify and include the sub-inhibitory concentrations used in the text for the fungal testing.
- e. I believe the authors identified 19 resistant isolates to fluconazole. Can they please speculate as to why only 12/25 were re-sensitised to fluconazole?
- f. Again, using this figure, it is difficult to ascertain the changes in gene expression, and no statistics are available to validate the statistical significance of the changes in expression observed. Further, p-values should be included in the accompanying results text section.
- g. The use of the abbreviations MICR, MICB, and MICA are confusing.
- h. The statement on lines 203-205 should be reconsidered as not all genes are markedly downregulation, and it appears very strain/isolate specific. These data should also be discussed further concerning how this impacts the hypothesis that rafoxanide works by downregulating beta-lactamase expression. This is not a universal observation, given the strain differences observed.
- i. What is the biological relevance of a 0.286-fold change in expression (line 208)? This number is minimal, and it is difficult to understand its significance as the text is currently written. Do the authors mean a log₂-fold of 0.2 or a 2.3-fold down-regulation? Though I note this would not align with their heatmap figure key. Again, I find this very confusing, and it needs to be clarified further or presented in an easier-to-interpret way.
- j. Is analysing the data using mean values across 50+ bacterial isolates of different genetic backgrounds appropriate?
- k. The text on lines 203-217 needs significant modification. Again, I think a stand-alone figure (i.e., bar graph) demonstrating the gene expression of treated and untreated cells should be shown with the appropriate statistical analysis.
- l. Why weren't any TCSs associated with antimicrobial resistance or virulence analysed via qPCR?

14. Figure 8 is difficult to interpret due to the many treatment groups shown. It would be easier for the reader to understand this data if it was split into separate graphs.

Further, the figure lacks any statistical analysis (though it is reported in the results text). Please add the statistical analysis directly to the figure and describe the statistical analysis conducted on this *in vivo* data (e.g., was this a log-rank test?).

- a. Please indicate when the animals were dosed with the therapeutics on the figure or in the figure legend.
 - b. Please clarify the treatment regime on lines 455-456.
 - c. What does the 72 h time frame refer to? How many treatment doses were given during this 72 h?
 - d. Please consider relabeling the Figure title. This is not survival probability. Survival analysis would be more appropriate.
 - e. Please report the dose challenge as total CFU, not CFU/mL
 - f. Please report the isolate names of each bacterial or fungal strain used in this experiment
 - g. Typo in the figure key change 'DR' to 'MDR'.
 - h. Please report the exact concentrations of each antibiotic or antifungal in mg/kg.
 - i. Please label the figure axis (Survival (%) and Hours-post infection)
 - j. Correct typo in the x-axis (12 to 120)
 - k. Please fix the formatting of bacterial and fungal names in the figure key.
 - l. Was a pre-infection survival analysis investigated for each MDR isolate (bacterial and fungal)? How was the dosing determined? This could be reported in the materials and methods section.
 - m. Please provide a brief explanation of the rationale for using both IV and IP infection routes. What model of infection was being investigated?
15. Regarding the *in-silico* modelling as it is currently written the rationale for this experiment is unclear. Please provide context as to why CitA was selected for this *in silico* analysis and how it relates to antimicrobial resistance. Or, was it chosen because it represents a typical two-component system?
- a. Please indicate in the figure legend that multiple rafoxanide compounds are modelled.
 - b. Please consider discussing how this *in silico* modelling relates to the fungal mechanism of resistance re-sensitisation/synergy.

Other Comments

Line 63-70; entirely vague and poorly cited.

Line 71; rewrite for clarity

Line 77; repertoire reports? Unclear

Line 82-84; provide context for 14-alpha-demethylase (i.e., its role in ergosterol synthesis).

Line 95-97; rewrite for clarity

Line 102; can you provide an example of a well-studied TCC inhibitor as an antimicrobial? Or maybe some additional information, if any inhibitor, has progressed into clinical trials for use as an antimicrobial?

Line 103; check reference 20; is this the correct reference?

Line 101-104; repetitive, edit.

Line 113; fix bacterial naming

Line 115; 'inhibition of HK protein' vague, rewrite

Line 123; were these animal ear swabs?

Line 129; do you mean multi-drug resistant (MDR) rather than multi-virulent?

Line 129; virulence genes' were the presence of these genes confirmed via sequencing? Or was the characterisation based on AMR susceptibility? Clarify. Also, please be consistent with using virulence genes or MDR genes.

Line 131-132; unclear; please define what MDR means (i.e., resistance to more than three drugs) if this differs from multi virulence (as used in line 129).

Lines 135-136; switching between 'most effective drugs' and 'susceptibility patterns' is confusing. What does this mean? Most effective drugs presumably refer to antibiotics which showed the lowest percentage of resistant isolates of the strains examined.

Line 159; MDE should read MDR

Line 181; typo mL, ML

Line 184; typo in 'Figure'. Figure S1 is missing.

Line 192, please add units

Line 193-196; rewrite for clarity

Line 285-287; is this an issue if you use sub-therapeutic doses of rafxanide in combination with an antibiotic?

Line 287-289; rewrite for clarity

Line 298-301; I find the logic here confusing, as the authors had previously attributed this effect to the downregulation of beta-lactamase enzymes and are now discussing the role of histidine kinases

Line 302-303; this citation does not suggest that rafxanide targets histidine kinases but indicates that the salicylanilide class of compounds can do so; please adjust the text in the discussion accordingly.

Line 306-307; please provide further evidence for this statement

Line 310; remove bracket '(B'

Line 312; no evidence was provided to support the statement that rafxanide binds HK receptors in fungi.

Line 373; add strain details, please

Dear Professor Doctor/ Editor-in-Chief of Microbiology Spectrum Journal

The manuscript ID: Spectrum02679-22

Title: Therapeutic switching of rafoxanide: A new approach to fight resistant bacteria and fungi

Many thanks for the Reviewers' comments and the opportunity to further revise the paper. We would like to thank the reviewers for their raised and thorough comments. The corrections requested by the reviewers have been done point by point as shown in the revision form. Hopefully, our revised manuscript meets the expectations of you and the reviewers and be considered for publication in Microbiology Spectrum Journal.

Responses to the reviewer:

General Comments:

1. The abstract would benefit from further clarification. For example, on lines 37-39, what does 'resistance patterns' mean? It is not apparent from reading the abstract in isolation what this term refers to, likewise for 'associated percentage'?

Thank you for your considerable comment. The meaning of the resistance patterns was clarified in the abstract.

a. Abstract Line 38; please consider clarifying where these isolates originated (i.e., animal, and human clinical samples, environmental samples etc.).

Thank you for your vital comment. All required data were clarified in details in the revised manuscript.

b. It is unclear in the abstract how the authors moved from examining the expression of AMR-related genes (i.e., beta-lactamases) to focusing on TCSs/histidine kinases. The author's reasoning for doing so could be clarified if the rationale for selecting and focusing on rafoxanide is explained when the compound is introduced. Regarding this, more information should be provided in the introduction establishing the role of rafoxanide as an inhibitor of TCSs.

I applaud you for the well understanding of the rational for the selection of rafoxanide. As you know, rafoxanide has salicylanilide nucleus. Several reports confirmed the role of salicylanilide compounds to reduce the antimicrobial resistance. Therefore, we postulated the synergistic effect of rafoxanide with other antimicrobial drugs and we success to ensure this role through our *in vitro* (MIC and FIC values) and molecular (expression of resistance genes) aspects in addition to the *in vivo* and molecular docking studies. In accordance with a previous study, one of the accepted explanation for the therapeutic role of rafoxanide against resistant pathogens is the inhibitory effects on histidine kinase enzymes. In this context and in accordance with the reviewer point of view, we added more information in the introduction section about this issue.

c. Please edit Lines 46-47 in the abstract as there is no direct evidence provided in the manuscript to indicate that inhibition of histidine kinases protein activity occurs with

rafoxanide or that this plays a role in the mechanism of resistance reversal. The only data provided to support this hypothesis is in silico modelling. Further, no data is provided in the manuscript to indicate this is the mechanism in the fungal isolates examined.

This paragraph was edited in the revised manuscript to be more accurate.

d. On line 42 of the abstract, please clarify the units associated with '<2'.

Thank you for your comment. It was clarified.

e. On line 44 of the abstract, please consider providing more detail regarding 'improvement in the activity,' i.e., provide x-fold changes or FICI values.

Thank you for your comment. More details were provided.

2. The methods section of the manuscript describing antimicrobial susceptibility testing is unclear. For example, what media was used for the bacterial testing, presumably cation-adjusted Mueller Hinton broth? Please consider adding more detail. Similar comments for testing of the Fungi, were the CLSI guidelines followed?

Thank you for your obvious comment. More details were added.

a. Further, how were the results (i.e., breakpoints) interpreted, i.e., according to CLSI guidelines or EUCAST guidelines?

Thank you for your obvious comment. Interpretation of results was clarified.

b. Consider combining the susceptibility testing method section with the checkerboard section, as there is some overlap in the text.

Thank you for your comment. The overlap in the text in both sections were removed. Firstly, we determined the susceptibility of the recovered bacterial and fungal isolates against the common used antimicrobials using both Kirby-Bauer disc diffusion and broth microdilution methods. Secondly, we evaluated the antibacterial and antifungal activities of rafoxanide against the tested bacterial and fungal isolates alone via broth microdilution technique and in combination with the least effective antimicrobials via the checkerboard method. All these data were clarified in the revised manuscript.

3. In the applicable Figure legends, please describe how the heatmap and hierarchical clustering were performed and what software packages (with citations) were utilised.

Thank you for your additive comment. It was added in the statistical analysis in the material and methods section.

4. Please consider describing the gene function of *MOX* and *DHA* in the introduction concerning antibiotic resistance and provide citations. More information is required about these two genes and their roles (i.e., AmpC). Line 76 would be a good place for this brief clarification.

Thank you for your additive comment. The functions and roles of *MOX* and *DHA* genes in the resistance were added in the introduction section in the referred place recommended by the reviewer.

5. Can you please provide an accurate reference for the role of rafoxanide as a TCS agonist?

Is there direct evidence for this, or is this rationale based on data with other salicylanilides? This could be made very clear in the introduction to support the role of rafoxanide as a TCS agonist.

Thank you for your comment. The relevant reference was added in the revised version.

6. Is reference 24 correctly cited? Does this citation indeed provide direct evidence that rafoxanide is a TCS agonist; I can't seem to find this information in the cited paper. Please double-check all referencing is appropriate for statements made in the manuscript.

Thank you for your valuable comment. We apologize for this unintended mistake. We corrected this reference in the revised manuscript and replaced it with the correct references. The salicylanilide nucleus of rafoxanide has been confirmed as a TCSs antagonist, which inhibits the bacterial HK through enzymatic allosterism. The following is the salicylanilide nucleus.

As it was shown in the structure of rafoxanide, it contains the salicylanilide nucleus. Therefore, the activity of this drug may be attributed to this moiety.

7. Please consider editing the statement on lines 110-111. Do these citations provide direct evidence that rafoxanide alters the expression of beta-lactamase genes? Please double-check all referencing is appropriate.

Thank you for your comment. All the paragraph regarding the evidence that rafoxanide alters the expression of beta-lactamase genes was modified and accordingly, we added the correct citations.

8. Please consider splitting Figure 8 into three separate graphs (A, B, and C) for each microbe examined and please add the statistical analysis to the graphs in Figure 8. I have made other comments on this figure below.

Thank you for your comment. We already splitted Figure 8 into four separate graphs (A, B, C and D) with associated standard errors.

Specific Comments:

1. Figure S1. Can the authors provide bacterial viability data (i.e., CFU/mL?). Further, please provide a detailed figure legend and information about biological replicates.

Thank you for your additive comment. We provided viability data (CFU/mL) via constructing a new supplementary figure based on the reviewer comment. Moreover, we added a detailed figure legend with information about biological replicates. Moreover, all data regarding growth curve analysis were added in the materials and methods section.

2. Based on previously established literature, rafoxanide has been proposed to inhibit oxidative phosphorylation. Is there a reason this was this not investigated further as a central mechanism behind rafoxanide-mediated resistance reversal? Please consider clarifying this in the manuscript. Doing so might help the reader understand your focused approach.

Thank you for your additive comment. There is no evidence in any previous studies proving that rafoxanide can act on microbial oxidative phosphorylation in contrast to their antiparasitic activities; meanwhile, several other targets and pathways were postulated as a central mechanism. We clarify this point in the new revised version of our manuscript and we added these issues to discuss this point of view.

a. Further, other anthelmintic drugs (such as oxiclozanide related) are known to disrupt/alter the permeability of the Gram-negative cell wall. Again, the authors could provide a rationale for their narrow focus on beta-lactamase enzymes, while the literature suggests the authors could have investigated other hypotheses.

Thank you for your valuable comment. The disruption and alteration of membrane permeability is postulated as one of the several pathways, through which rafoxanide exerts its antimicrobial activities and this was well discussed in several studies, but this action cannot be taken as a central mechanism as it can cause the minimal damage to cause bacterial death. On the other hand, the decrease in the expression of beta-lactamase genes wasn't the pathway for rafoxanide antimicrobial activity, but it is only an indication for the synergistic action of rafoxanide when used with other β -lactam antibiotics only. The reduction in the expression of resistance and virulence genes can be directly correlated with the inhibition of histidine kinase protein and we tried to link the effect of rafoxanide on histidine kinase protein as a primary target with the expression of resistance genes. This was announced in the new revised version of our manuscript and we added the correspondent paragraphs to discuss this point of view.

3. Given the author's hypothesis that rafoxanide inhibits histidine kinases, why were no studies conducted to examine this effect directly? This gap could be addressed in the discussion and included in future work.

Thank you for your valued comment. In the new revised version, we discussed with numerous previous studies the direct effect of rafoxanide on histidine kinases and this issue was added also in the limitation section of this study to be taken into our consideration in the future researches.

4. On lines 181-184, please provide detailed information about what bacterial and fungal isolates were selected for this analysis.

Thank you for your comment. The required detailed information was provided.

a. Does the . MIC value etc., refer to . of the mean MIC or the measured MIC for the specific strain investigated?

Thank you for your comment. We meant mean MIC values as clarified in the revised manuscript.

5. The figure legend for Figure 1 is entirely unclear and requires more detail so the reader can complete a stand-alone analysis. For example, what exactly does the 'MDR pattern' refer to?

Thank you for your significant comment. We clarified what exactly the MDR pattern refers to in the figure legend.

a. Do the 60+40 isolates refer to 100 in total (for *E. coli*)? Or does it mean 40/60 (66%) isolates had an 'MDR pattern'? When examining the figure, it's not apparent; the authors can resolve this with further description in the figure legend.

Thank you for your substantial comment. It means that 40/60 (66%) isolates had MDR patterns and this was clarified in the figure legend. More description was added in the figure legend to be independently understood.

b. Is this 'MDR pattern' based on antimicrobial susceptibility data or the presence of resistance genes as determined by PCR etc.? I realise this is described on lines 370-371, but it should be clarified in the Figure legend text.

Thank you for your additive comment. The required data was clarified in the figure legend text.

c. Further, the authors frequently use full names, e.g., *Aspergillus fumigatus* or *A. fumigatus*. Consistency is required throughout the manuscript.

Thank you for your noticeable comment. We went through the whole manuscript and checked all bacterial and fungal species names for consistency. We mentioned only the full names for the first time with their abbreviations and then we mentioned the abbreviations only. Moreover, on all figures and their legends, we made the same corrections for consistency.

d. Fix formatting of *E-coli* to *E. coli*.

Thank you for your noticeable comment. All that was corrected as per the previous comment.

e. The associated results section for Figure 1 should highlight for the reader how many isolates were of animal or human origin and if this had any impact on the susceptibility profiles.

Thank you for your noticeable comment. All isolates' sources were added in the revised version of the manuscript.

6. I have similar comments regarding the legend of Figure 2 – more detail is required to allow stand-alone analysis.

Thank you for your obvious comment. More details were added as required.

a. Did the authors take, for example, all 40 *E. coli* MDR pattern isolates and individually test all 40 isolates against the described antibiotics? Was this conducted by micro broth dilution, in what media, and under what conditions for bacterial and fungal isolates? Please mention biological replicates and what threshold was used to determine resistance (i.e., EUCAST or CLSI breakpoints). The authors should include mention of this in the results section.

Thank you for your obvious comment. More information was added in the figure legend and in the material and methods section to make it clear for the readers.

b. Please fix various formatting issues in Figure 2. Italicise bacterial and fungal names, correct capitalisation and correct mixed use of full and abbreviated names.

Thank you for your careful referee. All these issues were revised and corrected.

c. Note that Figure 2A x-axis ends at 90%; Fig 2B ends at 100%.

Thank you for your noticeable comment. It was revised and uniformed.

7. In Figure 3 and the associated results text as it is currently written, the value of the hierarchical clustering within a heatmap (Figure 3) is unclear. Further, the authors should provide more supporting text regarding this analysis in the results section on lines 138-146.

Thank you for your informative comment. More information and supportive text were included in the new revised version. The required information were added in the revised manuscript.

a. Was there any correlation between species and where they were isolated from (human or animal) with their antimicrobial resistance profile? It appears that there are four major clusters shown; the authors could consider commenting on these clusters, what they represent, where they were isolated from and the resistance profiles.

Thank you for your additive comment. The answers of these question were added in the results and figure legends of the new revised version of our manuscript.

b. What method was used to perform the clustering and distance matrix? These details should be made clear in the figure legend.

Thank you for your comment. The method used to perform this dendrogram was introduced in the figure legend and in the material and methods section.

c. It is unclear why the heat map key was included when the authors only used two colours.

Thank you for your observable comment. We apologize for this unintended mistake; therefore, the key was corrected to include two colors only and the color code was introduced in the figure legend.

d. It is unclear what genotypic profile traits were used to generate this figure.

Thank you for your evident comment. In this figure, only phenotypic detection of antifungal resistance [presence/absence] were used to construct this figure; meanwhile, the genetic basis of the antifungal resistance was detected depending on the increasing in the expression of the resistance genes (*C. albicans* ERG11 and *A. fumigatus* cyp51A), which were existed in all tested isolates. More information about this point were incorporated in the figure legend.

8. Figure 4 and the results it represents are challenging to interpret. For example, the antibiotic resistance genes could be placed together. I found using red to highlight antibiotic resistance and the presence of an antibiotic resistance gene confusing.

Thank you for your apparent comment. The resistance genes were distinguished from the phenotypic resistance in the new revised version of our manuscript. The resistance genes were written in italic format and red colour and this was illustrated in the figure legend and more other details were also incorporated. We are sorry as we cannot collect

the resistance genes in one place as the distribution of the isolates and the related criteria were done using the R program. Meanwhile, we tried to avoid this confusion by changing the colour and the formatting of the resistance genes. Additionally more information were added in the figure legend and in the result section.

9. Move lines 166-170 to the antifungal activity section

Thank you for your obvious comment. This result was moved to the antifungal activity section.

a. MICs are generally not written with +SDs.

Thank you for your comment. We totally agree with your comment and we delete +SDs and write only the MIC range for the investigated isolates.

b. Further, was this mean MIC value across all strains examined or only 'MDR pattern' isolates? Please make this clear in the results section.

Thank you for your considerable comment. The mean MIC values were calculated across all strains examined from each species. This was cleared in the results section of the revised manuscript.

c. Please also clearly state the MIC values against the control ATCC strains and include this information in the text.

Thank you for your comment. We used control ATCC strains as positive and negative controls for PCR assays and we did not determine the MIC values against them.

d. This section would benefit enormously from including the MIC data for the strains examined in a Table in supplementary information or the main text. Likewise, the associated FICs should be included in this table.

Thank you for your comment. Supplementary tables 1 and 2 including the MIC data for the examined bacterial and fungal strains and the corresponding FICs were added in the new version of the revised manuscript.

10. These MIC values appear to be relatively low for rafoxanide against Gram-negative bacteria. For example, Domaloan *et al.* (10.1038/s41429-019-0186-8) demonstrated that rafoxanide had very high MICs against *E. coli* (>256 ug/mL), and a similar result was observed by Miro-Canturri *et al.* with rafoxanide MICs against *K. pneumoniae* of >256 ug/mL. The authors should consider adding some commentary around the differences in these observed sensitivities of the stand-alone activity of rafoxanide and contrast this with their data.

Thank you for your noticeable comment. Of note, the mean MIC values of rafoxanide against Gram-negative bacteria in this study were low in contrast to the previous publications as some *E. coli* isolates showed abnormal sensitivity to rafoxanide, which leads to decreases in the mean values of rafoxanide MIC. Therefore and in accordance to the previous reviewer comment, the mean values were deleted and the range of the MIC values was introduced. Regarding the differences in these data and those of previous studies, we added more information in this point in the discussion section.

11. Please clarify the statement on line 166, 'measuring the expression levels'; in the context of a section describing molecular detection by PCR, it is confusing.

Thank you for your valuable comment. This statement was rephrased and additional related information was added in the introduction section.

a. Did you use RT-qPCR to detect the gene expression of ERG11 and cyp51A? Or was some analysis of the presence of these two fungal genes also conducted by PCR only? This section is very confusing as you are no longer talking about molecular detection but rather the gene expression between antifungal (i.e., azole) treated and untreated fungal cells. These data could be moved to another more appropriate section of the manuscript. Consider moving this to lines 211-213.

Thank you for this advice, but our point of view about these results is that we used RT-qPCR firstly for the molecular detection of fluconazole resistance among resistant fungal isolates. These resistant isolates were then selected for evaluating the effect of rafoxanide. Although the two paragraphs were about the detection of the expression levels of fungal resistance genes, we cannot merge them in one paragraph as the first part was about the detection of resistance (upregulation of the related genes) and the other paragraph was about the effect of rafoxanide, which convert resistant isolates to sensitive (downregulation of the related genes).

b. Further, providing detailed information on the level of upregulation observed with appropriate statistical analysis would be beneficial.

Thank you for this comment. Other supportive information about the level of upregulation for each bacterial and fungal isolate was introduced in 2 supplementary tables. Moreover, statistical analysis for the results of gene expression was also provided in the results section with an additional supplementary figure.

c. Interpreting these data from Figure 5 alone is very difficult; a bar graph with relative gene expression levels would be easier to analyse and allow you to show the relevant analysis.

Thank you for your comment. It is very impossible to construct a bar graph for the relative gene expression for all investigated isolates (100 bacterial and 25 fungal isolates); meanwhile, a bar graph depending on the mean values of fold changes for each type of bacterial or fungal isolates was constructed and provided as a supplementary figure in the new version of the revised manuscript. The response to the previous comment can solve the difficulty in reading this figure as we added 2 supplementary tables, which contain all values about the levels of genes expression for all examined isolates.

d. Please provide additional discussion on the biological relevance of a 0.218-0.7-fold change in gene expression levels.

Thank you for your considerable comment. More information, which reflects the biological relevance of a 0.218- 0.7-fold change in gene expression levels was added in both introduction and discussion sections.

12. It is unclear to me why the authors did not investigate changes in gene expression levels to compare A) rafoxanide treated cells, B) antibiotic or antifungal treated cells, and C) rafoxanide+antifungal (or antibiotic) treated cells. Suppose you were to compare the gene expression levels under all three conditions relative to untreated controls. In that case, you could potentially observe more significant changes in the gene expression levels and have more robust data to validate your hypothesis.

Indeed, measuring the expression levels of the investigated genes in the treated isolates with both antimicrobial drugs and the tested compound could potentially add more robust data, but; unfortunately, this point was not taken in our account during designing the methodology of our research. Therefore, we will take this point seriously into consideration in the future researches. For achieving our aim, we measured in the

current study the expression levels of the examined resistance genes on the same isolates before and after treating with rafoxanide, which provides us with the findings that rafoxanide can convert the susceptibility patterns of the examined isolates from resistant to sensitive.

a. Additionally, some microorganisms produce beta-lactamase enzymes in a regulated way (i.e., in the presence of the beta-lactam antibiotic). Including the beta-lactam antibiotic in a treatment group might ensure that the beta lactamase genes are upregulated, allowing the authors to investigate the effect of rafoxanide under more appropriate experimental conditions.

Thank you for this valuable comment. We totally agree with this information; therefore, we added this point in the limitation of this study to give more attention about this point to other researchers.

b. Further, if the authors believe rafoxanide-mediated changes to betalactamase expression play a key role in overcoming beta-lactam resistance, could they consider the importance of investigating the gene expression of specific regulators of beta-lactamase expression?

Thank you for this considerable comment. Upon the provided new data in the introduction and discussion sections about the role of rafoxanide on the downregulation of beta-lactamase genes, it was documented that histidine kinase enzyme binds with specific receptor on the promoter of β -lactamase gene at specific sequences (cre/blr-tag: TTCACnnnnnnTTCAC) leading to upregulation of this gene with subsequent increase in the production of beta-lactamase enzyme. In our report, we approved this role on the protein level via molecular docking studies (binding of rafoxanide with bacterial histidine kinases) and on the gene expression level (fold change in the expression levels of β -lactamase genes). Meanwhile, we did not measure the fold change in the expression of the specific promoter. Therefore, we will take this point into our consideration in future researches and we added this in the limitation section.

c. Can the authors please clarify if any of the beta-lactamase enzymes investigated in the pathogens described are under the control of two component regulons? Some additional text in the discussion regarding this would be helpful.

Thank you for this comment. We clarified this point in details in the introduction and discussion sections.

13. Figure 6 is tough to interpret

a. Firstly, it is unclear what the exact MICs are for all strains examined against cefotaxime. By using the coloured legend, only an approximation of the MIC can be interpreted by the reader. I think this MIC data is better reported in a Table with the exact values.

Thank you for this valuable comment. Two supplementary tables with detailed MIC data were supplied in the new version of our manuscript.

b. It is unclear which breakpoint value the authors use for cefotaxime; this should be described. For example, EUCAST sets a resistance breakpoint of ≥ 2 ug/mL (for indications other than meningitis). Please add this information to the results text.

Thank you for this comment. The break point values for cefotaxime and other additional information were added in the methodology.

c. The legend for Figure 6 should indicate the sub-MIC value, and if this was consistent across all bacterial strains, please clarify what was used. i.e., . MIC? This should also be made clear in the accompanying results text. Again, the use of a table would solve this issue.

As suggested before, we added two supplementary tables illustrating the exact MIC values in addition to sub-MIC values of rafoxanide for all investigated bacterial and fungal strains and we also added all these values on the new figures as per the reviewer comment for more clearance of data.

d. Lines 196-200 are difficult to understand. Do the authors mean that 12/25 fungal isolates were re-sensitized to fluconazole (below MIC breakpoints) in the presence of sub-inhibitory rafoxanide? At the same time, other strains demonstrated a reduction in the MICs, but this was not clinically relevant (i.e., not below MIC breakpoints?). Please clarify and include the sub-inhibitory concentrations used in the text for the fungal testing.

Thank you for your comment and we applaud you for your well understanding of this paragraph. The MIC values were variable between the tested strains. Therefore, we cannot add the exact MIC values; meanwhile, we used the sub-MIC term, which is equivalent to 0.5 MIC values for each strain. The exact MIC values were supplied in the added supplementary tables. Moreover, this paragraph was rephrased to be easily understood.

e. I believe the authors identified 19 resistant isolates to fluconazole. Can they please speculate as to why only 12/25 were re-sensitized to fluconazole?

Thank you for your observable comment. We refined this point and rephrased this paragraph to be clearer.

f. Again, using this figure, it is difficult to ascertain the changes in gene expression, and no statistics are available to validate the statistical significance of the changes in expression observed. Further, *p*-values should be included in the accompanying results text section.

Thank you for your valuable comment. The added supplementary table and the new figure could solve this difficulty. Moreover, we added the statistical analysis (*p*-values) for the results of gene expression in the accompanying results with an additional supplementary figure.

g. The use of the abbreviations MICR, MICB, and MICA are confusing.

Thank you for your valuable comment. We tried to decrease this confusion using more precise abbreviations for MIC-R, MIC-B, and MIC-A.

h. The statement on lines 203-205 should be reconsidered as not all genes are markedly downregulation, and it appears very strain/isolate specific. These data should also be discussed further concerning how this impacts the hypothesis that rafoxanide works by downregulating beta-lactamase expression. This is not a universal observation, given the strain differences observed.

Thank you for your comment. All bacterial and fungal resistance genes in this study were downregulated upon exposure to sub-MICs of rafoxanide, which are specific for each species. In accordance with the reviewer comment, this will be clearly observed in the added supplementary table and the new figure. Additionally, more information about this point was added in the discussion section.

i. What is the biological relevance of a 0.286-fold change in expression (line 208)? This number is minimal, and it is difficult to understand its significance as the text is currently written. Do the authors mean a log₂-fold of 0.2 or a 2.3-fold down-regulation? Though I note this would not align with their heatmap figure key. Again, I find this very confusing, and it needs to be clarified further or presented in an easier-to-interpret way.

Thank you for your vital comment. We meant here by this value the mean fold change in the expression of *bla*_{CTX-M-1} gene in the investigated *E. coli* isolates after exposure to sub-MICs of rafoxanide, which reflects the downregulation of this gene after treatment of the isolates with rafoxanide. The actual fold change values for all examined isolates ranged from 0.128 to 0.392 and this was cleared in the revised manuscript. Regarding the heatmap figure key, figure 6 was replaced by another one with clearer key correspondent to fold change values of the isolates. Moreover, all these values were illustrated on the new heatmap in the revised manuscript.

j. Is analysing the data using mean values across 50+ bacterial isolates of different genetic backgrounds appropriate?

Thank you for your considerable comment. We totally agree with your opinion, but we determined the mean values across these isolates as there were no great differences between their values and the SEs reflect these results.

k. The text on lines 203-217 needs significant modification. Again, I think a stand-alone figure (i.e., bar graph) demonstrating the gene expression of treated and untreated cells should be shown with the appropriate statistical analysis.

Thank you for your considerable comment. We totally agree with your suggestion and accordingly, we added a new supplementary figure demonstrating the mean fold changes for the investigated resistance genes of treated and untreated isolates with the appropriate statistical analysis.

l. Why weren't any TCSs associated with antimicrobial resistance or virulence analysed via qPCR?

Thank you for your significant comment. The action of rafoxanide on the TCSs was approved in different studies and that was explained in the new version of introduction in the revised manuscript. This action is through the inhibition of histidine kinase protein (not the gene). Therefore, evaluating the action of rafoxanide must be carried out on the protein level. The qPCR was used in our study to evaluate the net results of the effect of rafoxanide on the histidine kinase enzyme, which was proven via its downregulatory effect on resistance genes. Furthermore, the action of rafoxanide on the protein level was evaluated using the molecular docking study that approved our results. In this context, we added more information in the discussion section about the action of the rafoxanide on the protein level, which affects the expression of resistance genes.

14. Figure 8 is difficult to interpret due to the many treatment groups shown. It would be easier for the reader to understand this data if it was split into separate graphs. Further, the figure lacks any statistical analysis (though it is reported in the results text). Please add the statistical analysis directly to the figure and describe the statistical analysis conducted on this *in vivo* data (e.g., was this a log-rank test?).

Thank you for your comment. Figure 8 was split into four separate graphs as recommended by the reviewer. Regarding to the statistical analysis, SEs were added within the graph bars.

a. Please indicate when the animals were dosed with the therapeutics on the figure or in the figure legend.

Thank you for your comment. The animals were dosed after 4 h inoculation with bacterial and fungal isolates every 24 h for 3 successive days (72 h) and this note was added in the figure legend.

b. Please clarify the treatment regime on lines 455-456.

Thank you for your comment. It was clarified.

c. What does the 72 h time frame refer to? How many treatment doses were given during this 72 h?

Thank you for your comment. We apologize for this uncompleted data. We added the dose regime in details. We meant by 72 h that we inject the drug every 24 h for 3 successive days [3 doses] as recommended previously (Miró-Canturri et al., 2020).

Miró-Canturri A, Ayerbe-Algaba R, Villodres ÁR, Pachón J, Smani Y. 2020 Repositioning rafoxanide to treat Gram-negative bacilli infections. *J Antimicrob Chemother* 75:1895–1905.

d. Please consider relabeling the Figure title. This is not survival probability. Survival analysis would be more appropriate.

Thank you for your comment. It was relabeled and adjusted in accordance with your point of view.

e. Please report the dose challenge as total CFU, not CFU/mL

Thank you for your comment. It was added in both CFU and CFU/mL.

f. Please report the isolate names of each bacterial or fungal strain used in this experiment

Thank you for your comment. The isolates' names were indicated in the methodology; meanwhile, the isolates codes were added in the results.

g. Typo in the figure key change 'DR' to 'MDR'.

Thank you for your comment. It was revised and corrected.

h. Please report the exact concentrations of each antibiotic or antifungal in mg/kg.

Thank you for your comment. Our point of view is that the exact concentrations of each antibiotic or antifungal depend on their MIC values against the used isolates in this experiment, which are considered as results of our work. Therefore, to avoid this interruption, we added the exact concentrations in the result section of this experiment according to their MIC values against the investigated isolates.

i. Please label the figure axis (Survival (%) and Hours-post infection)

Thank you for your comment. The figure axis was labeled as requested.

j. Correct typo in the x-axis (12 to 120)

Thank you for your comment. It was corrected.

k. Please fix the formatting of bacterial and fungal names in the figure key.

Thank you for your comment. The names were fixed.

l. Was a pre-infection survival analysis investigated for each MDR isolate (bacterial and fungal)? How was the dosing determined? This could be reported in the materials and methods section.

Thank you for your valuable comment. From our point of view, the pre-infection survival analysis wasn't essential in this situation, especially in the presence of both negative and positive control groups. The doses of MDR isolates were determined based on previous papers those were cited on our revised manuscript and more information in this point were added in the new version of our revised manuscript.

m. Please provide a brief explanation of the rationale for using both IV and IP infection routes. What model of infection was being investigated?

Thank you for your valuable comment. We apologize for providing very small brief for the methodology of the *in vivo* study as we tried to concise our methodology depending on previous publications providing the citations for each step. We completely agree with your comment; therefore, we added the possible details to this test in the new version of our revised manuscript. Regarding both IV and IP infection routes and in accordance with previous publications in this field, the route of bacterial infection to create observed morbidity and morbidity was the IP route; meanwhile, the fungal infection must be injected through the IV route.

15. Regarding the *in-silico* modelling as it is currently written the rationale for this experiment is unclear. Please provide context as to why CitA was selected for this *in silico* analysis and how it relates to antimicrobial resistance. Or, was it chosen because it represents a typical two-component system?

Thank you for your important comment. Two-component system (TCS), which is comprised of a histidine kinase (HK) and a response regulator (RR) is a common mechanism, whereby bacteria can sense a range of stimuli and make an appropriate adaptive response. Citrate binding to the extracytoplasmic sensing domain of histidine kinase CitA triggers a contraction of the domain resulting in a pull on the C-terminal β -strand (<https://doi.org/10.1073/pnas.1620286114> and doi: 10.3389/fchem.2022.866392). So, histidine kinase CitA is regarded as a promising target to overcome microbial resistance in the past decades (doi: 10.3389/fchem.2022.866392). Consequently, histidine kinase CitA was selected for our docking studies.

a. Please indicate in the figure legend that multiple rafoxanide compounds are modelled.

Thank you for your vital comment. We would like to declare that modeling was carried out on only one compound, which is rafoxanide (anthelmintic drug).

b. Please consider discussing how this *in silico* modelling relates to the fungal mechanism of resistance re-sensitisation/synergy.

Thank you for your vital comment. Modeling was conducted as well on sensory fungal histidine kinase revealing the inhibitory potential and binding interactions of rafoxanide.

Other Comments:

Line 63-70; entirely vague and poorly cited.

Thank you for your comment. This paragraph was completely changed.

Line 71; rewrite for clarity

Thank you for your comment. It was rephrased to be clearer.

Line 77; repertoire reports? Unclear

Thank you for your comment. It was replaced by several researches.

Line 82-84; provide context for 14-alpha-demethylase (i.e., its role in ergosterol synthesis).

Thank you for your comment. Its role was added.

Line 95-97; rewrite for clarity

Thank you for your comment. It was rephrased to be clearer.

Line 102; can you provide an example of a well-studied TCC inhibitor as an antimicrobial?

Or maybe some additional information, if any inhibitor, has progressed into clinical trials for use as an antimicrobial?

Thank you for your comment. Examples of well-studied TCC inhibitors as antimicrobials were provided.

Line 103; check reference 20; is this the correct reference?

Thank you for your comment. We apologize for this wrong citation. We replaced it with the correct ones.

Line 101-104; repetitive, edit.

Thank you for your comment. It was edited to avoid repetition.

Line 113; fix bacterial naming

Thank you for your comment. All bacterial species were written in a complete manner in the first time they mentioned and then they were mentioned abbreviated.

Line 115; 'inhibition of HK protein' vague, rewrite

Thank you for your comment. It was rewritten.

Line 123, were these animal ear swabs?

Thank you for your comment. The 100 ear swabs were collected from patients suffering from otitis media and dogs presenting clinical otitis (50 each). Therefore, this sentence was cleared.

Line 129; do you mean multi-drug resistant (MDR) rather than multi-virulent?

Line 129; virulence genes' were the presence of these genes confirmed via sequencing?
Or

was the characterisation based on AMR susceptibility? Clarify. Also, please be consistent with using virulence genes or MDR genes.

Thank you for your comment. We apologize for this unintended mistake. We already mean MDR, so we deleted this sentence in this position and mentioned these

phenomenon percentages in details in their correct places under antimicrobial susceptibility patterns title.

Line 131-132; unclear; please define what MDR means (i.e., resistance to more than three drugs) if this differs from multi virulence (as used in line 129).

Thank you for your valuable comment and worthy observation. The MDR meaning was added and we already deleted the unintended mistake regarding multi virulence and defined MDR in its correct position.

Lines 135-136; switching between 'most effective drugs' and 'susceptibility patterns' is confusing. What does this mean? Most effective drugs presumably refer to antibiotics, which showed the lowest percentage of resistant isolates of the strains examined.

Thank you for your comment. All these issues were clearly defined.

Line 159; MDE should read MDR

Thank you for your comment. It was corrected.

Line 181; typo mL, ML

Thank you for your comment. It was corrected.

Line 184; typo in 'Figure'. Figure S1 is missing.

Thank you for your comment. We apologize for this accidental mistake. The supplementary figure 1 was corrected and provided as per the reviewer comments.

Line 192, please add units

Thank you for your comment. The units were added.

Line 193-196; rewrite for clarity

Thank you for your comment. It was rewritten for clarity.

Line285-287; is this an issue if you use sub-therapeutic doses of rafoxanide in combination with an antibiotic?

Thank you for your comment. This issue was resolved when using sub-therapeutic doses of rafoxanide in combination with antibiotics, where promising results for these combinations were recorded. These results were clarified in the results and discussion sections.

Line 287-289; rewrite for clarity

Thank you for your comment. It was rewritten for clarity.

Line 298-301; I find the logic here confusing, as the authors had previously attributed this effect to the downregulation of beta-lactamase enzymes and are now discussing the role of histidine kinases

Thank you for your comment. For clarification, for β -lactam antibiotics, TCS is involved in the upregulation and activation of β -lactamase enzyme. The membrane-associated histidine kinases act as transmembrane signals, which induce the production of β -lactamase enzymes through the upregulation of β -lactamase genes. This correlation between TCS including histidine kinase enzyme and the production of β -lactamase enzymes was confirmed previously in several researches. All that was clarified in the discussion section of the revised manuscript.

Line 302-303; this citation does not suggest that rafoxanide targets histidine kinases but indicates that the salicylanilide class of compounds can do so; please adjust the text in the discussion accordingly.

Thank you for your comment. All these issues were corrected targeting the halogenated salicylanilide and the text in the discussion was adjusted accordingly.

Line 306-307; please provide further evidence for this statement

Thank you for your comment. This statement was evidenced through referring to the binding scores for the binding affinity of rafoxanide with bacterial and fungal HK proteins as proved in our study and in the additional work we made according to the reviewer comments.

Line 310; remove bracket '(B'

Thank you for your comment. It was removed.

Line 312; no evidence was provided to support the statement that rafoxanide binds HK receptors in fungi.

Thank you for your comment. This sentence was corrected to include HK receptors in fungal cells as provided from the additional work we made as per the reviewer comments.

Line 373; add strain details, please

Thank you for your comment. The ATTC number for the susceptible strain was added.

May 9, 2023

Dr. Nada Hashim
University of Gezira Faculty of Medicine
Wad Medani
Sudan

Re: Spectrum02679-22R1 (Therapeutic switching of rafoxanide: A new approach to fight resistant bacteria and fungi)

Dear Dr. Nada Hashim:

Link Not Available

Sincerely,

Francisco Uzal

Journals Department
Reviewer comments:

Reviewer #2 (Comments for the Author):

The review of your study named Therapeutic switching of rafoxanide: A new approach to fight resistant bacteria and fungi has been completed. First of all, congratulations for this hard work. However, the text and bibliography were as long as the compilation articles. You can shorten the text so that the integrity of the subject is not disturbed, the work will be easier for readers to read. Also, it would be appropriate for you to correct the two points I mentioned below.

1. The expression "ug/ml" in line 379 should be corrected as "µg/ml".
2. The expression "helminthes" in various places in the text should be corrected as "helminths".

I wish good work.

Staff Comments:

Preparing Revision Guidelines

Please return the manuscript within 60 days; if you cannot complete the modification within this time period, please contact me. If you do not wish to modify the manuscript and prefer to submit it to another journal, please notify me of your decision immediately so that the manuscript may be formally withdrawn from consideration by Microbiology Spectrum.

Dear Professor Doctor/ Editor-in-Chief of Microbiology Spectrum Journal

The manuscript ID: Spectrum02679-22

Title: Therapeutic switching of rafoxanide: A new approach to fight resistant bacteria and fungi

Many thanks for the Reviewers' comments and the opportunity to further revise the paper. We would like to thank the reviewers for their raised and thorough comments. The corrections requested by the reviewers have been done point by point as shown in the revision form. Hopefully, our revised manuscript meets the expectations of you and the reviewers and be considered for publication in Microbiology Spectrum Journal.

Responses to the reviewer:

Reviewer #2 (Comments for the Author):

The review of your study named Therapeutic switching of rafoxanide: A new approach to fight resistant bacteria and fungi has been completed.

We feel great thanks for your professional review work on our article and for your precise and considerable summary for our study. We have certainly benefited from your insightful suggestions.

First of all, congratulations for this hard work. However, the text and bibliography were as long as the compilation articles. You can shorten the text so that the integrity of the subject is not disturbed, the work will be easier for readers to read.

Thank you for your additive comment. We go precisely through the manuscript to shorten the text and therefore to meet your suggestions. Several paragraphs in the introduction and discussion were merged together in order to not affect the content. The entire text was reduced from 11388 to 9709 words and from 99 to 76 bibliography and the total number of pages was also reduced from 45 to 39 pages.

Also, it would be appropriate for you to correct the two points I mentioned below.

1. The expression "ug/ml" in line 379 should be corrected as "µg/ml".

2. The expression "helminthes" in various places in the text should be corrected as "helminths".

Thanks for these valuable observations. They were both corrected.

June 5, 2023

Dr. Nada Hashim
University of Gezira Faculty of Medicine
Wad Medani
Sudan

Re: Spectrum02679-22R2 (Therapeutic switching of rafoxanide: A new approach to fight resistant bacteria and fungi)

Dear Dr. Nada Hashim:

Your manuscript has been accepted, and I am forwarding it to the ASM Journals Department for publication. You will be notified when your proofs are ready to be viewed.

Sincerely,

Francisco Uzal
Editor, Microbiology Spectrum
